# What Limits Virtual Agent Application?
# OmniBench: A Scalable Multi-Dimensional Benchmark for Essential Virtual Agent Capabilities

**Wendong Bu** [1 2 *]  **Yang Wu** [2 *]  **Qifan Yu** [1 *]  **Minghe Gao** [1]  **Bingchen Miao** [1]  **Zhenkui Zhang** [1]  **Kaihang Pan** [1]
**Yunfei Li** [2]  **Mengze Li** [3]  **Wei Ji** [4]  **Juncheng Li** [1 ✉]  **Siliang Tang** [1]  **Yueting Zhuang** [1]

## Abstract

As multimodal large language models (MLLMs) advance, MLLM-based virtual agents have demonstrated remarkable performance. However, existing benchmarks face significant limitations, including uncontrollable task complexity, extensive manual annotation with limited scenarios, and a lack of multidimensional evaluation. In response to these challenges, we introduce **OmniBench**, a self-generating, cross-platform, graph-based benchmark with an automated pipeline for synthesizing tasks of controllable complexity through subtask composition. To evaluate the diverse capabilities of virtual agents on the graph, we further present **OmniEval**, a multidimensional evaluation framework that includes subtask-level evaluation, graph-based metrics, and comprehensive tests across 10 capabilities. Our synthesized dataset contains 36k graph-structured tasks across 20 scenarios, achieving a 91% human acceptance rate. Training on our graph-structured data shows that it can more efficiently guide agents compared to manually annotated data. We conduct multidimensional evaluations for various open-source and closed-source models, revealing their performance across various capabilities and paving the way for future advancements. Our project is available at https://omni-bench.github.io/.

## 1. Introduction

With the development of MLLMs (Fei et al., 2024c; Wu et al., 2024a), recent MLLM-based virtual agents (Xu et al., 2024b; Gao et al., 2024a; Miao et al., 2025) have demonstrated promising performance in web navigation (Shen et al., 2024), mobile device control (Ge et al., 2024), and computer interaction (Hu et al., 2024). To explore real-world values of visual agents, current research mainly evaluates their performance based on offline trajectory similarity with human demonstrations (Rawles et al., 2024; Lu et al., 2024; Deng et al., 2024) or by using expert-crafted functions in interactive online environments (Xie et al., 2024; Zhou et al., 2023; Yao et al., 2022).

However, these two types of benchmarks mentioned above still have notable limitations: **1) Uncontrollable and fixed task complexity.** Existing benchmarks typically propose tasks entirely rather than progressively with fine-grained guidance, which results in uncontrollable and fixed task complexity. Uncontrollable task complexity makes it hard to design fine-grained test data for various capabilities, while fixed complexity makes it challenging for benchmarks to keep up with agents' growing capabilities. **2) Extensive manual labor and limited task scenarios.** The existing benchmarks rely on manual annotations to synthesize demonstration trajectories or evaluation functions, making the cost of designing benchmarks unaffordable and hindering the expansion of scale. Moreover, the annotated data with a limited amount is influenced by human prior experience, making it difficult to cover comprehensive scenarios. **3) Absence of multidimensional evaluation**: Existing benchmarks commonly evaluate agents based on the final state of tasks, lacking an evaluation of the intermediate steps in task execution. Additionally, the various capabilities required by virtual agents to complete tasks (*e.g.*, planning, instruction understanding, etc.) cannot be quantified by coarse task success rates, failing to provide sufficient feedback for potential future improvements. **In summary**, an ideal benchmark should include not only diverse task scenarios with controllable complexity, but also a comprehensive evaluation across multiple dimensions.

To cost-effectively construct diverse task scenarios with complexity at multiple granularities for comprehensive agent evaluation, we propose a novel self-generating, graph-

---

*Equal contribution  [1]Zhejiang University, Hangzhou, China [2]Ant Group, Hangzhou, China [3]The Hong Kong University of Science and Technology, Hong Kong SAR, China [4]Nanjing University, Nanjing, China. Correspondence to: Juncheng Li <junchengli@zju.edu.cn>.

*Proceedings of the 42nd International Conference on Machine Learning*, Vancouver, Canada. PMLR 267, 2025. Copyright 2025 by the author(s).

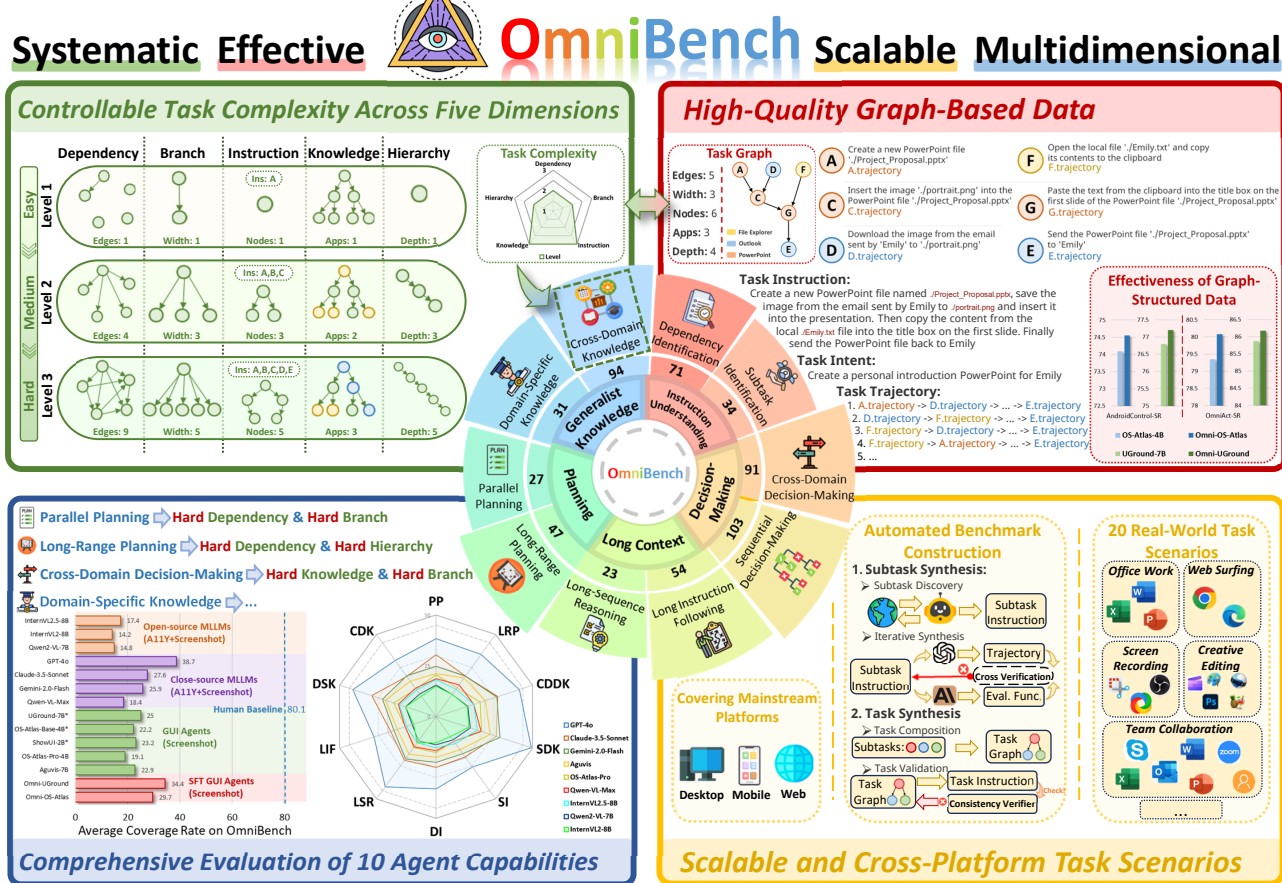

*Figure 1.* Overview of OmniBench, a systematic benchmark with five-dimensional task complexity and bottom-up automatic task synthesis for generating structured task graphs. It evaluates ten virtual agent capabilities using high-quality graph-based data, ensuring scalable and realistic task evaluation.

based benchmark, **OmniBench**. It dynamically synthesizes tasks with controllable complexity based on a bottom-up pipeline. OmniBench spans five fundamental types of task complexity to construct 10 evaluation dimensions (see Figure 1). Test tasks across these dimensions are categorized based on combinations of complexity types. For example, a long-range planning test task typically exhibits higher dependency complexity and hierarchy complexity. OmniBench consists of 36k high-quality graph-structured tasks across 20 distinct scenarios (*e.g.* image editing, video editing) derived from its self-generating framework, with the task scale being 40x larger than most environment-based benchmarks, as shown in Table 1. This automated process opens up the possibility of scaling up virtual agent evaluation in a low-resource manner. Therefore, OmniBench facilitates the easy construction of agent benchmarks on desktop, mobile, and web platforms, as shown in Table 1. For multidimensional task synthesis, we take motivations from the DAG topology (Xu et al., 2024a) to design a bottom-up pipeline. Specifically, we define five fundamental task complexities and **synthesize tasks with controllable complexity by constraining the DAG composition process**. Notably, we extracted task intents to guide this process to

avoid composing meaningless tasks (*e.g.*, opening a food delivery app and immediately closing it). We further incorporate quality control modules to optimize subtasks and ensure semantic alignment. In this way, we cost-effectively synthesize high-quality graph-structured tasks, significantly **broadening task scenarios without requiring human annotations**.

Moreover, due to the inherent complexity of conducting comprehensive and fine-grained evaluations of agents, we propose a graph-based multidimensional evaluation framework, **OmniEval**. In contrast to previous coarse-grained evaluation methods, we introduce a graph-based evaluator that leverages subtask-level evaluation functions synthesized in OmniBench. Specifically, we design two novel fine-grained metrics to evaluate agents' performance on graph-structured tasks and their alignment with human logic. Based on OmniBench, we comprehensively evaluate 12 virtual agents, including both open-source and proprietary models, across all 10 capability dimensions as shown in Figure 1, **fully revealing the capability boundaries and providing concrete directions for future improvement**.

The performance comparison between various models pro-

*Table 1.* Comparison of virtual agent benchmarks across environment, task, and evaluation dimensions. Unlike previous benchmarks, OmniBench features automatic task composition, five-dimensional task complexity, and a 10-capability evaluation framework.

| | Environment | | | Task | | | | | | | | | Evaluation | | | |
| --- | --- | --- | --- | --- | --- | --- | --- | --- | --- | --- | --- | --- | --- | --- | --- | --- |
| | Interactive | Real-World | Platform | # Instance | # Compl. Dimen. | Dyna. Scale | Intent | # Scenario | Demo. Traj. | Construction | Instruction Level | # Cap. Dimen. | Eval. Level | # Eval. Func. | Evaluation Strategy |
| AitW (Rawles et al., 2024) | ✗ | ✗ | 📱 | 30378 | - | ✗ | ✗ | 5 | ✓ | Manual Annotation | High & Low | 1 | Task | - | Trajectory-based |
| Mind2Web (Deng et al., 2024) | ✗ | ✗ | 🌐 | 2350 | - | ✗ | ✗ | 5 | ✓ | Manual Annotation | High | 1 | Task | - | Trajectory-based |
| MoTIF (Burns et al., 2022) | ✗ | ✗ | 📱 | 756 | - | ✗ | ✗ | - | ✓ | Manual Annotation | High & Low | 1 | Task | - | Trajectory-based |
| OmniACT (Kapoor et al., 2025) | ✗ | ✗ | 💻🌐 | 9802 | - | ✗ | ✗ | 6 | ✓ | Manual Annotation | Low | 1 | Task | - | Trajectory-based |
| GUI Odyssey (Lu et al., 2024) | ✗ | ✗ | 📱🌐 | 7735 | - | ✗ | ✗ | 6 | ✓ | Manual Annotation | Low | 1 | Task | - | Trajectory-based |
| WebArena (Zhou et al., 2023) | ✓ | ✗ | 🌐 | 812 | - | ✗ | ✗ | 4 | ✗ | Manual Annotation | Low | 1 | Task | 5 | Result-based |
| VisualWebArena (Koh et al., 2024) | ✓ | ✗ | 🌐 | 910 | 2 | ✗ | ✗ | 3 | ✗ | Manual Annotation | Low | 1 | Task | 6 | Result-based |
| OSWorld (Xie et al., 2024) | ✓ | ✓ | 💻 | 369 | - | ✗ | ✗ | 5 | ✗ | Manual Annotation | Low | 1 | Task | 134 | Result-based |
| Spider2-V (Cao et al., 2024) | ✓ | ✓ | 💻🌐 | 494 | 1 | ✗ | ✗ | 7 | ✗ | Manual Annotation | High & Low | 1 | Task | 151 | Result-based |
| CRAB (Xu et al., 2024a) | ✓ | ✓ | 💻📱 | 100 | - | ✓ | ✗ | - | ✗ | Manual Composition | Low | 1 | Subtask | 59 | Graph-based |
| OmniBench (Ours) | ✓ | ✓ | 💻📱🌐 | 36076 | 5 | ✓ | ✓ | 20 | ✓ | Automatic Composition & Human Verification | High & Low | 10 | Subtask | 255 | Graph-based |

vides valuable findings for future virtual agent application. Specifically: **1) Existing agents struggle to handle graph-structured tasks**. Compared to tasks with linear structures, the agents fall significantly short when facing graph-structured tasks, with even GPT-4o achieving only 20.5% performance, while humans can reach 80.1%. **2) Task intents are crucial for task planning**. Incorporating task intents into the prompt offers a plug-and-play improvement to planning performance, with an average increase from 23.4% to 28.9%. Similarly, using task intents in fine-tuning data improves planning performance from 30.5% to 31.9%. **3) Mainstream agents are sensitive to expression order in task instructions**. We observe significant performance fluctuations in existing agents when altering the expression order of task instructions. In contrast, agents fine-tuned with graph-structured trajectories exhibit more stable performance.

Furthermore, we fine-tune two open-source agents with distinct architectures on synthesized graph-structured task trajectories. As shown in Figure 1, both agents exhibit performance improvements on AndroidControl and Omni-Act. Compared to agents trained on manually annotated datasets, our agents achieve better performance across diverse benchmarks, benefiting from reasoning-rich trajectories that demonstrate their broad applicability and strong potential.

## 2. Related Work

**Virtual Digital Agents.** With the development of MLLMs (Pan et al., 2025b; 2024b; Li et al., 2023a; Fei et al., 2024a), virtual agents have greatly improved task automation across platforms. CogAgent (Hong et al., 2024) introduced an 18B visual language model for GUI understanding, achieving state-of-the-art performance. SeeClick (Cheng et al., 2024) developed a vision-only model that interacts with GUIs via screenshots, eliminating the need for structured data. UGround (Gou et al., 2024) proposed a universal grounding model, accurately mapping GUI elements across platforms. Iris (Ge et al., 2024) enhances GUI automation by tackling challenges in complex digital environments. Evaluating these visual agents is crucial for real-world applications.

**Benchmarks for Virtual Agents.** Mainstream benchmarks

for virtual agents are generally categorized into two types: trajectory-based and result-based. Trajectory-based benchmarks (Rawles et al., 2024; Cheng et al., 2024; Deng et al., 2024) compare agent trajectories to human demonstrations but can be inaccurate due to the existence of multiple valid trajectories. Result-based benchmarks (Xie et al., 2024; Zhou et al., 2023; Cao et al., 2024) focus on the final state of the environment, overlooking the fine-grained evaluation of intermediate processes. More recently, some studies (Shen et al., 2023; Xu et al., 2024a) have introduced graph-based evaluations, which support both multiple feasible trajectories and the evaluation of intermediate processes. TASKBENCH (Shen et al., 2023) evaluates agents for task automation, but its simplistic metrics fail to fully utilize the potential of graph structures. CRAB (Xu et al., 2024a) evaluates agents using handcrafted graphs, but it lacks a systematic task analysis, limiting fine-grained capability assessment. Notably, to the best of our knowledge, **OmniBench** is the only scalable benchmark for virtual agents that defines composable task complexity using graphs to evaluate multiple essential capabilities.

## 3. OmniBench

OmniBench consists of 36k high-quality graph-structured tasks across 10 evaluation dimensions to simulate the way humans perceive the digital world, including planning, decision-making, etc. In this section, we first introduce task graphs of OmniBench and systematically define corresponding task complexities (Section 3.1). Then, we present the bottom-up data collection pipeline for controllably synthesizing tasks (Section 3.2). Additionally, we explain how to control the quality of the synthesized data (Section 3.3). Finally, we showcase the statistics of OmniBench (Section 3.4).

### 3.1. Task Complexity on Task Graph

In this section, we define task complexity on graphs, which is later constrained in Section 4.2 to construct test tasks for multidimensional capability. We propose a new complexity definition because existing benchmarks (Wang et al., 2024a; Cao et al., 2024; Koh et al., 2024) typically define task complexity based on the number of steps in human demonstrations. However, this approach has two limitations: 1) The inherent subjectivity of human demonstrations makes this definition unreliable; 2) It is one-dimensional and fails to capture the multifaceted complex-

*Figure 2.* Bottom-up task synthesis pipeline

*Table 2.* Complexity Dimensions and Their Corresponding Levels

| Complexity Dimension | Calculation | Easy | Medium | Hard |
|---|---|---|---|---|
| Dependency Complexity | Number of Edges | $\leq 1$ | $2 \sim 3$ | $\geq 4$ |
| Instruction Complexity | Number of Nodes | $\leq 2$ | $3 \sim 4$ | $\geq 5$ |
| Knowledge Complexity | Number of Application Categories | $\leq 1$ | $2 \sim 3$ | $\geq 4$ |
| Hierarchy Complexity | Depth | $\leq 2$ | $3 \sim 4$ | $\geq 5$ |
| Branch Complexity | Width | $\leq 2$ | $3 \sim 4$ | $\geq 5$ |

*Table 3.* Ablation study evaluating the impact of each quality control module on acceptability of the synthesized tasks.

| Cross-Verification | Intent Extraction | Consistency Validator | Human Acceptance |
|---|---|---|---|
| ✗ | ✗ | ✗ | 41.2% |
| ✗ | ✓ | ✓ | 61.2% |
| ✓ | ✗ | ✓ | 82.7% |
| ✓ | ✓ | ✗ | 86.5% |
| ✓ | ✓ | ✓ | 90.7% |

ity of real-world tasks. Inspired by previous works (Xu et al., 2024a; Shen et al., 2023) that represent tasks as graph structures, we introduced the concept of the task graph and systematically defined five fundamental task complexities on the task graph.

Specifically, we define a subtask as a smaller, independent task that contributes to completing a more complex task. Each subtask has input and output resources to constrain the dependencies between them. To formalize this, we assume each subtask as $s$ and define a task graph as $\mathcal{G} = \{S, R\}$, where $S = \{s_1, s_2, \ldots, s_n\}$ is the collection of subtasks, and $R$ is a set of relations $\{(s_a, s_b)\}$ indicating that subtask $s_b$ depends on subtask $s_a$ when the output resources of $s_a$ match the input resources of $s_b$.

After representing the task as a graph, a natural idea is to define task complexity based on the topology of the task graph. We systematically analyzed five characteristics of the graph and designed a corresponding five-dimensional task complexity. Specifically: **1) Dependency Complexity.** Since each edge in the task graph represents a dependency between subtasks, we define dependency complexity based on the number of edges. **2) Instruction Complexity.** The semantics of a task instruction are composed of all subtasks, with more subtasks leading to more complex instruction semantics. Therefore, we define instruction complexity based on the number of nodes. **3) Knowledge Complexity.** We categorize all 49 applications and define knowledge complexity based on the number of applications from different categories (*e.g.*, multimedia playback, productivity) in the task graph. The detailed categorization is provided in Appendix A.4. **4) Hierarchy Complexity.** The depth of the task graph represents the number of hierarchical levels in the task structure. Thus, we define hierarchy complexity based on the depth. **5) Branch Complexity.** A wider task graph indicates more branches that can be executed concurrently. Therefore, we define branch complexity based on the width. The classification criteria for three complexity levels are shown in Table 2.

### 3.2. Controllable Task Synthesis

Although we define five fundamental task complexities on the task graph, converting task instructions into a graph remains challenging. A straightforward idea is to directly top-down

decompose tasks into task graphs. However, this process is typically based on uncontrollable MLLMs or expensive manual efforts. Therefore, to effectively synthesize tasks with controllable complexity, we designed a bottom-up automated task synthesis pipeline, as shown in Figure 2.

**Overview.** The task synthesis pipeline we propose consists of four processes. First, we synthesize a series of simple subtask instructions from the explorable environment. Then, we iteratively synthesize subtask trajectories and evaluation functions. Next, the subtasks are combined into a task bottom-up. Finally, we validate the semantics of the tasks.

**Subtask Exploration.** We designed an environment containing 49 diverse applications, inspired by OSWorld (Xie et al., 2024), allowing advanced MLLMs to thoroughly explore each application to propose diverse and achievable subtasks. During exploration, documentation and example subtasks for each application are provided to help synthesis. To accurately synthesize the dependencies between subtasks, we provide a predefined resource list, which MLLMs use to determine the input and output resources for each subtask. The implementation details can be found in Appendix B.1.

**Iterative Synthesis.** We leverage advanced MLLMs to synthesize trajectories and evaluation functions of subtasks. For trajectories, our crafted prompts guide the MLLM to describe screenshots and output thoughts, improving inference on trajectories. For evaluation functions, we predefine 11 basic APIs to retrieve information such as clicked text, keyboard inputs, and file directory existence. Subsequently, we use Claude-3.5-Sonnet, which excels in the code domain, to compose these basic APIs into evaluation functions for subtasks. To improve the quality of synthetic data, we propose a novel cross-verification algorithm that iteratively refines the synthesis process. The implementation details can be found in Appendix B.2.

**Task Composition.** For high-quality subtask samples that pass cross-verification, we add them to the subtask pool. Directly bottom-up composing these subtasks into a task graph using input and output resources may result in tasks that lack a coherent core goal, such as "opening a food delivery app and immediately

closing it". To avoid synthesizing such low-quality tasks, we extract task intents for the composition scenarios from the subtask pool, such as "create a personal introduction PowerPoint for Emily," as shown in Figure 1. Each task intent involves a group of subtasks, which are then combined into a task graph using input and output resources. Since the bottom-up composition process is rule-based, The synthesis of the task graph is controllable, ensuring that tasks with controllable complexity can be synthesized. Implementation details are in Appendix B.3.

**Task Validation.** For the task graphs constructed through composition, we employ GPT-4o to summarize the task instruction based on the subtask instructions and the graph structure. However, such synthesized instructions may deviate from the original semantics of the task graph, such as losing the graph's nonlinear semantics and degenerating into a simple sequential task. To ensure the synthesis of high-quality graph-structured tasks, we designed a consistency validator to verify the semantic alignment between the task graph and the summarized task instruction. Specifically, GPT-4o determine the dependency for subtasks solely based on the task instruction. If the inferred dependencies match those in the task graph, the instruction passes validation; otherwise, the instruction needs to be re-summarized. Implementation details are in Appendix B.4.

### 3.3. Quality Control

Since the quality of graph-structured tasks is critical to the accurate evaluation of the virtual agents, we further introduce three designs to enhance the quality of synthesized data: a cross-verification mechanism, an intent extraction module, and a consistency validator. The cross-verification mechanism iteratively optimizes the demonstration trajectories and evaluation functions of subtasks, the intent extraction module ensures that the tasks have coherent goals, and the consistency validator aligns the semantics of the task graph and task instructions. We perform an ablation analysis of these three quality control methods, validating their effectiveness, as shown in Table 3. For each ablation shown in the table, we sampled 400 task graphs and calculated the average acceptance by three specially trained annotators. The experiment shows that removing any quality control module decreases human acceptance, with the removal of the cross-verification algorithm resulting in the largest drop to 61.2%.

### 3.4. OmniBench Statistics

OmniBench comprises a total of 36,076 task instances, spanning across 20 common interactive scenarios and involving 49 diverse applications, as illustrated in Appendix A.4. In Section 3.1, we introduced the five-dimensional complexity metrics for each task, with the distribution of different complexity levels for each dimension shown in Figure 3.

## 4. OmniEval

To comprehensively analyze the limited capabilities of existing agents, we further propose OmniEval, a graph-based multidi-

| Statistics | Value |
|---|---|
| **Total Tasks** | 36076 (100%) |
| - Network-dependent Real-world Tasks | 16614 (46.05%) |
| - Network-independent Local Tasks | 19462 (53.95%) |
| - Avg. Number of Used App Per Task | 2.21 |
| **Task Instruction** | |
| - Avg. Words of High-level Instruction | 51.7 |
| - Avg. Words of Low-level Instruction | 237.9 |
| Total Task Scenarios | 20 |
| Total Subtasks | 255 |

*Figure 3.* Statistics of OmniBench

mensional evaluation framework. In this section, we first introduce a graph-based evaluator with two novel metrics for fine-grained and diverse evaluation (Section 4.1). Then, we describe the construction of test tasks designed to evaluate 10 distinct capabilities by constraining task complexity (Section 4.2).

### 4.1. Graph-based Evaluator

Currently, most benchmarks still evaluate agents in a coarse-grained and unreasonable paradigm. Specifically, **result-based** evaluations (Xie et al., 2024; Zhou et al., 2023) consider whether the final environment state aligns with expectations, lacking fine-grained intermediate evaluation. As shown in Figure 4, although both trajectories ultimately fail to complete the task, the progress of trajectory 2 is superior to trajectory 1, and they should not simply be categorized as failures. While **trajectory-based** evaluations (Rawles et al., 2024; Lu et al., 2024) compare the agents' predicted actions to human demonstrations at each step, they overlook multiple feasible trajectories. As shown in Figure 4, both trajectories can accomplish the task, but trajectory 4 is deemed a failure because it does not align with the demonstration trajectory, which is unreasonable.

Considering the limitations of these two evaluation paradigms, we introduce a **graph-based** multi-metric evaluator inspired by previous research (Xu et al., 2024a), as shown in Figure 4. Specifically, we define three evaluation states for each node on the task graph: Completed, Evaluating, and Waiting. Initially, when the evaluator is set up, nodes with an in-degree of 0 are marked as Evaluating, while the remaining nodes are marked as Waiting. After the agent executes each action, it checks whether all Evaluating nodes have been completed. Once a node is completed, it is marked as Completed, and new Evaluating nodes are added according to the topological order. We set a maximum number of steps $N$, and if the agent does not complete any subtasks within $N$ steps, the entire task is considered a failure.

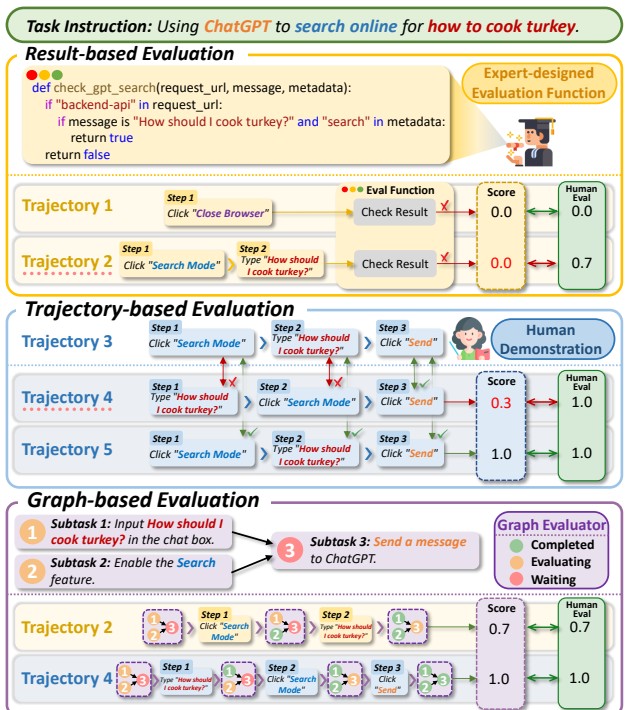

*Figure 4.* Comparison of mainstream virtual agent evaluation strategies with the evaluation strategy we propose.

Additionally, to fully leverage the potential of the graph-based evaluator, we have designed two novel graph-based metrics. Traditional metrics fail to evaluate intermediate processes and alignment with human operational logic. For example, common metrics such as **Success Rate (SR)** (Xie et al., 2024) focus on task outcomes rather than the process, while **Action Match Score (AMS)** (Li et al., 2020b) treats action sequences as strings and compares the similarity between human demonstrations and agent predictions, rather than their logical similarity. To comprehensively quantify agent performance on the task graph, we propose two novel metrics inspired by its topology. The **Coverage Rate (CR)** assesses agent progress on the task graph, while the **Logical Consistency (LC)** reflects the similarity in operational logic between agents and humans.

**Coverage Rate (CR).** It evaluates an agent's progress on a task graph by weighting subtasks based on their depth, where deeper subtasks are assigned higher weights due to their increased number of prerequisite subtasks. Referring to the relevant definitions in Section 3.1, let $d(s_i)$ denote the depth of subtask $s_i$. The weight $w(s_i)$ is:

$$w(s_i) = \frac{d(s_i)}{\sum_{j=1}^{n} d(s_j)}.$$

The Coverage Rate is then:

$$CR = \frac{\sum_{i=1}^{n} w(s_i) \cdot \mathbb{I}(s_i)}{\sum_{i=1}^{n} w(s_i)},$$

where $\mathbb{I}(s_i) = 1$ if subtask $s_i$ is completed, and 0 otherwise. This metric emphasizes deeper, more complex subtasks, providing a refined measure of agent performance.

**Logical Consistency (LC).** It quantifies the operational logic

similarity between agents and humans. This metric is motivated by the observation that humans prefer to complete all possible subtasks within an application before switching to another, unless necessary. It is computed as the ratio of the agent's Coherency Score (CS) to the maximum possible CS:

$$LC = \frac{CS_{agent}}{CS_{max}},$$

where $CS$ quantifies the coherence of the subtask sequence. For each pair of adjacent subtasks $(s_i, s_{i+1})$ in sequence, $CS$ increases by 1 if both subtasks belong to the same application. $CS_{agent}$ is the coherence score calculated from the executing subtask sequence, and $CS_{max}$ is the maximum possible coherence score calculated among all topological sequences.

## 4.2. Evaluation Strategy

Currently, there is limited discussion on the categorization of capabilities in the virtual agent field. The previous mainstream classifications (Lin et al., 2024; Gou et al., 2024) typically divided the capabilities into two simple categories: grounding and planning. However, this simple classification is quite coarse and does not take into account other essential and fine-grained capabilities for agents, such as decision-making and instruction understanding. To address this, we propose 10 fine-grained capabilities, derived from five categories that we consider essential, with each category contributing two capabilities for agents. Specific test tasks are constructed for each capability based on the combination of five complexity dimensions, as shown in Figure 1.

Taking long-range planning capability as an example, we categorize tasks with higher dependency complexity and hierarchy complexity as test tasks for this capability. This is because higher dependency complexity means the task involves more dependencies, requiring stronger planning capability. Meanwhile, higher hierarchy complexity indicates the task has deeper levels, which places higher demands on long-sequence processing capability. Therefore, we select tasks with dependency complexity and hierarchy complexity at the hard level as test tasks for long-sequence reasoning capability. For the test tasks of these 10 capabilities, we engaged professionally trained annotators to filter and construct high-quality test data. The specific definitions and corresponding explanations for the other 9 capabilities can be found in Appendix D.1.

## 5. Experiments

In this section, we first introduce the experimental setup (Section 5.1). Then, we comprehensively compare the differences in capabilities across various models on OmniBench, along with several key findings (Section 5.2). Finally, we provide an in-depth analysis of the reasons behind the poor performance and methods for performance improvement (Section 5.3).

### 5.1. Experimental Setup

**Settings.** We evaluate various models including MLLMs and Virtual Agents on OmniBench. For all virtual agents, we use the default prompt provided by each agent for inference, if available.

*Table 4.* Performance of models on OmniBench. For each capability, we use the CR metric on test tasks for quantification. Abbreviations adopted: PP for Parallel Planning; LRP for Long Range Planning; CDDK for Cross-Domain Decision-Making; SDK for Sequential Decision-Making; SI for Subtask Identification; DI for Dependency Identification; LSR for Long Sequence Reasoning; LIF for Long Instruction Following; DSK for Domain-Specific Knowledge; CDK for Cross-Domain Knowledge. An asterisk (*) indicates that the agent uses GPT-4o as the planner.

| | Overall | | Planning | | Decision-making | | Instruction Understanding | | Long Context | | Generalist Knowledge | |
|---|---|---|---|---|---|---|---|---|---|---|---|---|
| | *CR* | *LC* | *PP* | *LRP* | *CDDK* | *SDK* | *SI* | *DI* | *LSR* | *LIF* | *DSK* | *CDK* |
| Human | 80.1 | 92.8 | 80.1 | 76.9 | 91.9 | 93.0 | 69.1 | 72.1 | 79.5 | 66.1 | 89.4 | 71.5 |
| *Open-source Multimodal Large Language Models (A11Y+Screenshot)* | | | | | | | | | | | | |
| Qwen2-VL-7B (Wang et al., 2024b) | 14.8 | 9.0 | 15.5 | 13.5 | 16.5 | 17.8 | 14.1 | 13.8 | 14.7 | 12.4 | 15.8 | 13.8 |
| InternVL2-8B (Chen et al., 2024) | 14.2 | 13.0 | 15.0 | 13.5 | 16.2 | 16.9 | 12.1 | 12.9 | 15.8 | 11.7 | 15.8 | 12.4 |
| InternVL2.5-8B (Chen et al., 2024) | 17.4 | 18.8 | 18.2 | 16.7 | 19.6 | 21.5 | 16.4 | 15.3 | 15.8 | 15.4 | 19.0 | 16.3 |
| *Closed-source Multimodal Large Language Models (A11Y+Screenshot)* | | | | | | | | | | | | |
| Qwen-VL-Max (Bai et al., 2023) | 18.4 | 23.3 | 18.7 | 19.4 | 19.6 | 23.3 | 15.0 | 16.7 | 18.3 | 16.1 | 19.6 | 17.3 |
| Gemini-2.0-Flash | 25.9 | 38.0 | 24.8 | 24.6 | 31.5 | 33.2 | 22.5 | 22.5 | 25.7 | 21.9 | 27.8 | 24.8 |
| Claude-3.5-Sonnet | 27.6 | 35.0 | 30.5 | 24.7 | 32.0 | 31.3 | 24.5 | 25.0 | 26.6 | 23.5 | 33.4 | 24.5 |
| GPT-4o (Hurst et al., 2024) | **38.7** | **49.0** | **38.4** | **37.8** | **43.2** | **49.4** | **30.6** | **35.5** | **42.7** | **32.2** | **43.2** | **34.2** |
| *Visual Digital Agents (Screenshot)* | | | | | | | | | | | | |
| Aguvis-7B (Xu et al., 2024b) | 22.9 | 27.1 | 21.2 | 23.5 | 25.5 | 28.1 | 20.2 | 20.0 | 22.8 | 20.1 | 26.3 | 21.6 |
| OS-Atlas-Pro-4B (Wu et al., 2024b) | 19.1 | 23.9 | 20.6 | 17.6 | 22.9 | 23.6 | 15.0 | 17.7 | 18.7 | 15.9 | 22.0 | 16.8 |
| ShowUI-2B* (Lin et al., 2024) | 23.2 | 24.6 | 23.2 | 23.1 | 26.3 | 26.6 | 21.5 | 20.3 | 24.7 | 20.4 | 24.8 | 20.7 |
| OS-Atlas-Base-4B* (Wu et al., 2024b) | 22.2 | 23.8 | 23.2 | 21.9 | 26.2 | 25.6 | 19.4 | 19.5 | 23.5 | 20.0 | 23.4 | 19.3 |
| UGround-V1-7B* (Gou et al., 2024) | 25.0 | 26.3 | 25.7 | 25.1 | 30.6 | 31.4 | 21.5 | 21.3 | 24.8 | 21.3 | 27.2 | 21.5 |
| *Supervised Fine-Tuning Agents (Screenshot)* | | | | | | | | | | | | |
| Omni-OS-Atlas-Base-4B (Ours) | 29.7 | 30.1 | 24.2 | **33.0** | 34.9 | 35.3 | **28.7** | 24.2 | 27.9 | 26.5 | 33.8 | **28.2** |
| Omni-UGround-V1-7B (Ours) | **34.4** | **37.4** | **33.2** | 31.3 | **43.1** | **42.4** | 21.9 | **35.0** | **40.3** | **31.7** | **36.7** | 27.6 |

If models do not provide prompts for agent tasks, we use a unified prompt designed by us. We also report results trained on OmniBench data for some selected models. All experiments are conducted with NVIDIA A100 80G GPUs.

**Baselines.** We conduct a comprehensive evaluation of the four types of models as shown in Table 4. The specific details about the baselines can be found in Appendix D.2.2.

### 5.2. Main Results

**Alignment Between OmniEval and Human Evaluation.** Before delving deeper into the concrete evaluation results, we first compare the alignment between OmniEval and human evaluation for agent tasks. Specifically, we randomly sampled 50 trajectories from all models to calculate the correlation between OmniEval and human evaluation. Each trajectory was scored by two specially trained annotators, who referenced the task instructions to assign task completion scores and logical alignment scores from {0%, 10%, ..., 90%, 100%}. The final human evaluation score was determined by averaging the scores given by the two annotators. In Figure 5, we present the Pearson correlation between OmniEval and human evaluation. The results indicate

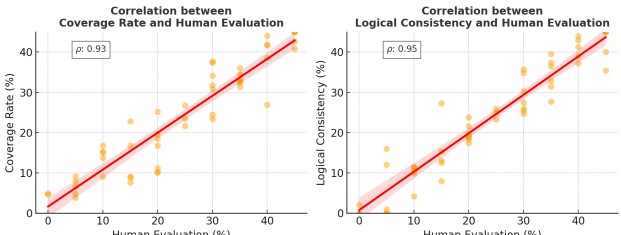

*Figure 5.* Correlation between Coverage Rate and Logical Consistency with Human Evaluation.

**Capability Boundaries of Mainstream Agents.** As detailed in Table 4, although advanced agents such as GPT-4o and our supervised fine-tuning models (e.g., Omni-UGround) demonstrate strong performance in overall metrics and in capabilities like planning and decision-making, clear limitations remain in Subtask Identification (SI) and Long Instruction Following (LIF). Specifically, even the strongest models only achieve 30.6 (GPT-4o) and 21.9 (Omni-UGround) in SI, and 32.2 and 31.7 respectively in LIF, which are significantly lower than the human baselines of 69.1 and 66.1, as shown in Figure 6. These results highlight a persistent difficulty in decomposing complex instructions and maintaining coherence over extended task flows. Compared to their performance in other capabilities, the results suggest that instruction understanding in long and semantically complex contexts remains a key bottleneck for current agents. Future improvements in agent performance will likely depend on more robust handling of multi-step semantics and long-context alignment.

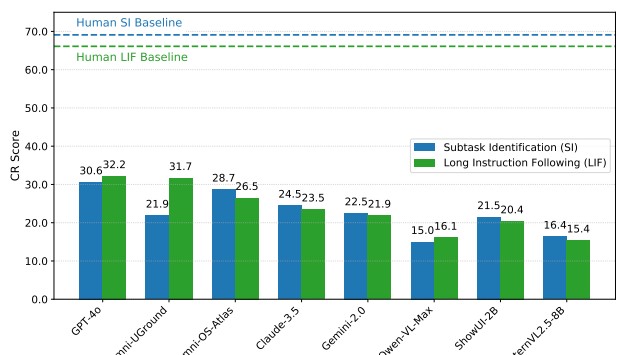

*Figure 6.* Comparison of model performance on Subtask Identification (SI) and Long Instruction Following (LIF).

*Table 5.* We compare the performance of different virtual agents on tasks with varying complexity levels. Medium and Hard levels show performance drops compared to the previous level, with downward arrows indicating the magnitude of decline.

| Agents | Dependency Comp. ↑ | | | Branch Comp. ↑ | | | Instruction Comp. ↑ | | | Knowledge Comp. ↑ | | | Hierarchy Comp. ↑ | | |
|---|---|---|---|---|---|---|---|---|---|---|---|---|---|---|---|
| | Easy | Medium | Hard | Easy | Medium | Hard | Easy | Medium | Hard | Easy | Medium | Hard | Easy | Medium | Hard |
| Aguvis-7B (Xu et al., 2024b) | 32.8 | 27.6 ↓5.2 | 24.3 ↓3.3 | 41.2 | 36.8 ↓4.4 | 30.6 ↓6.2 | 49.5 | 36.9 ↓12.6 | 25.3 ↓11.6 | 38.4 | 32.5 ↓5.9 | 27.6 ↓4.9 | 37.9 | 33.6 ↓4.3 | 29.7 ↓3.9 |
| OS-Atlas-Pro-7B (Wu et al., 2024b) | 32.3 | 26.8 ↓5.5 | 23.7 ↓3.1 | 39.1 | 31.0 ↓8.1 | 25.4 ↓5.6 | 44.3 | 34.8 ↓9.5 | 21.8 ↓13.0 | 33.9 | 28.4 ↓5.5 | 24.3 ↓4.1 | 34.5 | 28.1 ↓6.4 | 25.6 ↓2.5 |
| ShowUI-2B* (Lin et al., 2024) | 34.0 | 28.3 ↓5.7 | 25.6 ↓2.7 | 41.3 | 32.7 ↓8.6 | 28.2 ↓4.5 | 45.9 | 36.6 ↓9.3 | 25.4 ↓11.2 | 37.8 | 32.6 ↓5.2 | 27.4 ↓5.2 | 37.6 | 32.0 ↓5.6 | 28.1 ↓3.9 |
| OS-Atlas-Base-4B* (Wu et al., 2024b) | 32.7 | 29.1 ↓3.6 | 24.9 ↓4.2 | 35.2 | 32.4 ↓2.8 | 27.6 ↓4.8 | 48.2 | 37.5 ↓10.7 | 26.7 ↓10.8 | 39.1 | 34.2 ↓4.9 | 28.9 ↓5.3 | 43.1 | 38.4 ↓4.7 | 33.2 ↓5.2 |
| UGround-7B* (Gou et al., 2024) | 34.1 | 30.0 ↓4.1 | 27.1 ↓2.9 | 44.6 | 38.3 ↓6.3 | 32.4 ↓5.9 | 53.0 | 39.4 ↓13.6 | 27.2 ↓12.2 | 42.3 | 36.4 ↓5.9 | 32.6 ↓3.8 | 35.7 | 28.8 ↓6.9 | 25.5 ↓3.3 |

**Challenges in Handling Graph-structured Tasks.** We compared the performance differences of each model on chain-structured tasks and graph-structured tasks. To eliminate the influence of other factors, we utilized OmniBench's controllable task synthesis mechanism to construct a set of chain-structured and graph-structured tasks, each with the same number of nodes and edges. All tasks belong to the same knowledge domain and share the same level of knowledge complexity. As shown in Figure 7, GPT-4o (with A11Y), the most advanced agent, achieves only 20.5% accuracy on graph-structured tasks, far below the human performance at 80.1%. This phenomenon can be attributed to the fact that most existing agents are predominantly fine-tuned on chain-structured tasks, which may result in their tendency to interpret graph-structured tasks as linear. Such misinterpretation can significantly impair the agents' capability to accurately identify the dependency relationships between subtasks, ultimately leading to task execution failures.

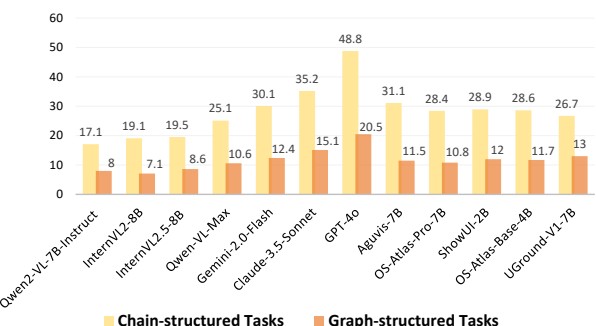

*Figure 7.* Performance comparison of various models on chain-structured tasks and graph-structured tasks.

*Table 6.* Evaluation results on AndroidControl benchmark. **Bold** values indicate the best performance across all baselines.

| Models | AndroidControl-Low | | | AndroidControl-High | | |
|---|---|---|---|---|---|---|
| | Type | Grounding | SR | Type | Grounding | SR |
| InternVL-2-4B (Chen et al., 2024) | 90.94 | 84.05 | 80.10 | 84.09 | 72.73 | 66.72 |
| Qwen2-VL-7B (Wang et al., 2024b) | 91.94 | 86.50 | 82.56 | 83.83 | 77.68 | 69.72 |
| SeeClick (Cheng et al., 2024) | 93.00 | 73.42 | 75.00 | 82.94 | 62.87 | 59.11 |
| OS-Atlas-4B (Wu et al., 2024b) | 91.92 | 83.76 | 80.64 | 84.69 | 73.79 | 67.54 |
| UGround-7B-V1 (Gou et al., 2024) | 92.15 | 87.17 | 83.29 | 84.72 | 78.85 | 70.31 |
| Omni-OS-Atlas-4B (Ours) | **92.49** | 83.51 | 81.38 | 84.86 | 73.81 | 67.71 |
| Omni-UGround-7B-V1 (Ours) | 92.37 | **87.24** | **83.57** | **84.89** | **78.97** | **70.83** |

*Table 7.* Evaluation results on OmniAct benchmark. **Bold** values indicate the best performance across all baselines.

| Models | OmniAct-Web | | | OmniAct-Desktop | | |
|---|---|---|---|---|---|---|
| | Type | Grounding | SR | Type | Grounding | SR |
| InternVL-2-4B (Chen et al., 2024) | 47.51 | 51.34 | 24.39 | 67.00 | 44.47 | 29.80 |
| Qwen2-VL-7B (Wang et al., 2024b) | 89.22 | 85.94 | 78.58 | 96.27 | 94.52 | 91.77 |
| SeeClick (Cheng et al., 2024) | 86.98 | 75.48 | 68.59 | 96.79 | 70.22 | 72.59 |
| OS-Atlas-4B (Wu et al., 2024b) | 88.56 | 82.00 | 73.91 | 96.51 | 85.53 | 84.78 |
| UGround-7B-V1 (Gou et al., 2024) | 90.16 | 86.98 | 79.85 | 97.13 | 94.79 | 91.89 |
| Omni-OS-Atlas-4B (Ours) | 89.96 | 82.74 | 74.62 | 97.64 | 86.37 | 85.53 |
| Omni-UGround-7B-V1 (Ours) | **91.24** | **87.35** | **80.24** | **97.93** | **95.21** | **92.10** |

### 5.3. In-Depth Analysis

**Performance Differences Across Complexity Levels.** As shown in Table 5, we analyze agent performance across tasks grouped by complexity levels: Easy, Medium, and Hard. Unsurprisingly, all agents exhibit significant performance drops as task complexity increases, with an average decrease of 6.19 points. This trend is consistent across all five dimensions. Such systematic degradation in harder cases confirms OmniBench's effectiveness in scaling task difficulty. Furthermore, performance on hard tasks may serve as a more accurate indicator of an agent's expert capabilities than average scores, revealing the upper bounds of its potential.

**Effectiveness of Graph-Structured Task Trajectories.** We follow the training details provided in the OS-Atlas paper and adopt the same experimental setup to train our backbone models: OS-Atlas-4B and UGround-7B-V1. As shown in Table 6 and Table 7, while agents such as OS-Atlas and UGround, which are pretrained on GUI grounding tasks, exhibit strong GUI understanding capabilities, their limited planning capability hinders their performance in complex action reasoning. In contrast, the high-quality multi-step navigation dataset synthesized by OmniBench significantly enhances the model's capability to make decisions regarding action types, thereby improving the success rate in GUI navigation. Specifically, Omni-OS-Atlas-4B achieves an average success rate improvement of 0.46 points on AndroidControl and 0.73 points on OmniAct, while Omni-UGround-7B-V1 achieves improvements of 0.4 points on AndroidControl and 0.3 points on OmniAct.

**Sensitivity to Expression Order in Task Instructions.** We define the impact of textual order on the model as its instruction sensitivity, conducting experiments with standard deviation as the metric. We construct 10 specially designed test tasks, each associated with three task instructions that are semantically identical (based on the same task graph) but differ in textual order. As shown in Table 8, the original MLLMs tend to be less sensitive to instruction variations, but perform poorly overall. Though fine-tuning them on navigation tasks enhances the performance, it also compromises the models' robustness to instructions. OS-Atlas-Pro and Aguvis exhibit significantly higher sensitivity, with an average increase of 8.21 points. Moreover, after incorporating graph-structured task samples from OmniBench into fine-tuning, the models' performance is further improved while largely preserving their robustness. Omni-OS-Atlas and Omni-Aguvis exhibit reduced sensitivity, with an average reduction of 7.91 points. This indicates that the diverse and structured task trajectories from OmniBench can help models better recognize complex dependencies embedded in task instructions, improving their overall stability and performance.

**The Effect of Task Intent on Planning.** We design two exper-

*Table 8.* Average sensitivity across different models.

| Models | Backbone | Avg. Sensitivity ↓ |
|---|---|---|
| Human | - | 1.95 |
| InternVL2-4B (Chen et al., 2024) | InternVL2-4B | 2.97 |
| OS-Atlas-Pro (Wu et al., 2024b) | InternVL2-4B | 9.07 ↑6.1 |
| Omni-OS-Atlas (Ours) | InternVL2-4B | 3.49 ↓5.58 |
| Qwen2-VL-7B (Wang et al., 2024b) | Qwen2-VL-7B | 2.58 |
| Aguvis (Xu et al., 2024b) | Qwen2-VL-7B | 12.9 ↑10.32 |
| Omni-Aguvis (Ours) | Qwen2-VL-7B | 2.67 ↓10.23 |

iments to explore the applicability of task intent to both open-source and closed-source models. **1) For open-source models**, we conduct a comparative experiment using two separate datasets to fine-tune OS-Atlas-Base-4B and UGround-V1-7B. One dataset includes task intent, while the other does not. As shown in Table 9, incorporating task intent in the training data significantly improves the model's planning performance on OmniBench. Specifically, OS-Atlas-Base-4B improves its overall planning score from 28.6 to 30.3, an increase of 1.7 points, while UGround-V1-7B improves from 32.3 to 33.5, gaining 1.2 points. **2) For closed-source models**, we use Qwen-VL-Max, Gemini-2.0-Flash, Claude-3.5-Sonnet, and GPT-4o as planners, with UGround-V1-7B serving as the grounding models. As shown in Table 9, all closed-source models exhibit improved planning performance with the inclusion of task intent. GPT-4o shows the most significant improvement, with its overall score rising from 25.4 to 34.3, a gain of 8.9 points. Claude-3.5-Sonnet increases by 5.4 points, Gemini-2.0-Flash by 4.9 points, and Qwen-VL-Max by 2.6 points. This indicates that closed-source models can enhance their planning through this plug-and-play approach.

*Table 9.* The effect of task intent on the planning of open-source versus closed-source models.

| Models | Parallel Planning ↑ | Long-Range Planning ↑ | Overall ↑ |
|---|---|---|---|
| *Open-source Multimodal Large Language Models* | | | |
| Omni-OS-Atlas-Base-4B | 24.2 | 33.0 | 28.6 |
| + intent tuning | 25.7 ↑1.5 | 34.9 ↑1.9 | 30.3 ↑1.7 |
| Omni-UGround-V1-7B | 33.2 | 31.3 | 32.3 |
| + intent tuning | 34.4 ↑1.2 | 32.6 ↑1.3 | 33.5 ↑1.2 |
| *Closed-source Multimodal Large Language Models* | | | |
| Qwen-VL-Max | 21.9 | 20.8 | 21.4 |
| + intent prompt | 24.5 ↑2.6 | 23.5 ↑2.7 | 24.0 ↑2.6 |
| Gemini-2.0-Flash | 23.1 | 22.7 | 22.9 |
| + intent prompt | 28.9 ↑5.8 | 26.7 ↑4.0 | 27.8 ↑4.9 |
| Claude-3.5-Sonnet | 24.2 | 23.7 | 24.0 |
| + intent prompt | 30.6 ↑6.4 | 28.1 ↑4.4 | 29.4 ↑5.4 |
| GPT-4o | 25.7 | 25.1 | 25.4 |
| + intent prompt | 32.9 ↑7.2 | 35.7 ↑10.6 | 34.3 ↑8.9 |

**Failure Analysis.** In this section, we delve into the analysis of errors encountered during the OmniBench evaluation. This analysis aims not only to identify the current shortcomings of the agents but also to inform future improvements in their design and training. We carefully examine 100 randomly sampled error instances for each model from the OmniBench evaluation. These instances are analyzed by expert annotators who identify the root causes of mispredictions based on their knowledge. Specifically, there are five types of errors: **1) Instruction Understanding.** We observe that 23% of the failures are due to the agent's misunderstanding of the instructions. For example, it overlooks the final

save file operation requested in the image editing instruction. **2) Lack of Knowledge.** We find that 21% of the failures are caused by the agent's lack of knowledge about the target application, such as being unfamiliar with how to create a reference list in Zotero. **3) Environmental Error.** We observe that 3% of the failures result from environmental interference, such as network delays. **4) Grounding Error.** We find that 17% of the failures are due to the model's lack of grounding ability, meaning the agent knows the target to click next but locates it in the wrong position. **5) Hallucinatory Success.** Finally, 36% of the failures occur when the agent incorrectly assumes the task is complete, which may stem from its weak contextual memory capabilities. The distribution of these errors is shown in Figure 8.

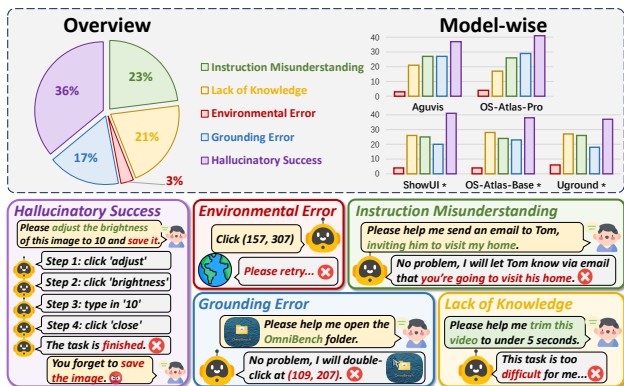

*Figure 8.* Distribution of five major errors in 100 failure instances for each model. An asterisk (*) indicates that the agent uses GPT-4o as the planner.

## 6. Conclusion

In conclusion, we introduced **OmniBench**, a graph-based benchmark that addresses the limitations of existing evaluation frameworks by enabling controllable task complexity through automated subtask composition. Along with **OmniEval**, a multidimensional evaluation framework, we evaluate virtual agents across 10 capabilities. Our results show that training on this data improves agent generalization, and our evaluations provide valuable insights into the strengths and areas for improvement in MLLM-based virtual agents.

**Acknowledgement.** This work was supported by the NSFC (62272411), the Fundamental Research Funds for the Central Universities (226-2025-00017), the Key R&D Projects in Zhejiang Province (No. 2024C01106, 2025C01030), Ningbo Yongjiang Talent Introduction Programme (2024A-401-G), the Zhejiang NSF (LRG25F020001), Ant Group.

## Impact Statement

This paper presents work whose goal is to advance the field of Machine Learning. There are many potential societal consequences of our work, none which we feel must be specifically highlighted here.

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

# Table of Contents in Appendix

# A. Environment Setup

OmniBench conducts agent evaluations across three categories of virtual environments: Desktop, Mobile, and Web. Its automated data collection pipeline makes it easy to extend the benchmark to additional environments with minimal effort. In the following section, we use the desktop environment as a case study to illustrate the environment design in OmniBench in more detail.

## A.1. Environment Infrastructure

Inspired by OSWorld (Xie et al., 2024), we design an interactive Windows-based environment using a virtual machine to support GUI agent evaluation. The environment runs Windows 11 as the guest operating system and uses VMware Workstation 17 Pro (version 17.5.1) as the virtualization platform. This setup enables high compatibility with real-world desktop applications while maintaining full control over the execution environment. The virtual machine allows us to simulate user interactions such as mouse clicks, keyboard input, and file operations, which are essential for GUI agents. It also supports real-time observation and logging of system states, facilitating fine-grained analysis and reproducibility of agent behavior. All environments are initialized from a snapshot to ensure consistent starting conditions for each evaluation episode.

## A.2. Observation Space

In OmniBench, the observation space is designed to ensure comprehensive evaluation of GUI agents by capturing both visual and structural aspects of desktop environments. It comprises two complementary modalities: screen captures and accessibility trees. This dual-modality approach reflects the varying grounding capabilities of different agent architectures. For example, agents that have been specifically trained on GUI environments often possess strong grounding abilities and can rely on screen captures alone. In contrast, MLLMs typically lack specialized pretraining for GUI understanding, and therefore benefit significantly from the semantic and structural information provided by the accessibility tree. By supporting both modalities, OmniBench enables fair and informative evaluation across a wide range of agents, ensuring robust evaluation under diverse UI layouts and application contexts.

## A.3. Action Space

In OmniBench, the action space consists of three core types of user interactions that an agent can perform. These actions, summarized in Table 10, enable the agent to effectively interact with graphical user interfaces across a wide range of applications.

*Table 10.* Summary of action types in the desktop environment of OmniBench.

| Action | Description |
|---|---|
| click_input | Simulates mouse clicks on UI control elements. Supports configurable mouse buttons (left, right, middle, x) and can perform both single and double clicks. Commonly used for selecting items, activating controls, or opening folders. |
| wheel_mouse_input | Scrolls vertically using the mouse wheel. Useful when target controls are not immediately visible. The scroll direction and distance are adjustable, allowing the agent to navigate long content or lists. |
| keyboard_input | Simulates keyboard input for typing text, pressing keys, or invoking shortcuts (e.g., Ctrl+C, Enter). Enables fine-grained control over application behavior and supports both functional and textual input. |

## A.4. Categorization of Applications

The 49 applications in the environment are categorized into 12 distinct groups based on their functionality: Social Communication, Multimedia Playback, Multimedia Editing, Office, Utility Tools, Programming, System Management, Web Browsing, Screen Capture, Task Management, Note Management, and Lifestyle. The specific applications belonging to each category are listed in Table 11.

*Table 11.* Categorization of applications in the environment.

| Category | Applications |
|---|---|
| Social Communication (4) | Zoom Workplace, Skype, People, Mail |
| Multimedia Playback (4) | Media Player, Spotify, Photos, TuneIn |
| Multimedia Editing (6) | Adobe Photoshop Express, Microsoft Clipchamp, paint.net, Openshot, Handbrake, Paint |
| Office (3) | Word, PowerPoint, Excel |
| Utility Tools (10) | Calculator, 7-Zip, PDF24, Power Automate, Wikipedia, BreeZip, Maps, Calendar, Zotero, DeepL |
| Programming (3) | Visual Studio Code, Cursor, Windows PowerShell ISE |
| System Management (4) | File Explorer, Settings, Control Panel, Microsoft Store |
| Web Browsing (2) | Google Chrome, Microsoft Edge |
| Screen Capture (4) | Record Screen, Snipping Tool, OBS Studio, ShareX |
| Task Management (3) | Microsoft To Do, Todoist, Notion |
| Note management (4) | Evernote, OneNote, Sticky Notes, Sticky Notes (New) |
| Lifestyle (2) | Recipe Keeper, paisa |

## A.5. Definition of Scenarios

As shown in Table 11, we categorize the 49 applications into 12 groups. Based on these application categories, we define 12 corresponding task scenarios. For example, applications in the "Multimedia Playback" category correspond to the "Media Viewing" scenario. We then combine these 12 basic task scenarios to form 7 more fine-grained task scenarios. For instance, the "Screen Recording" scenario can be combined with the "Creative Editing" scenario to form a more detailed scenario called "Screen Recording Editing." Through such combinations, we obtain a total of 19 task scenarios. The remaining tasks are grouped into an additional scenario, resulting in a final set of 20 task scenarios, as illustrated in Figure 9.

# B. Data Collection

OmniBench introduces a bottom-up pipeline for automatic task synthesis, as illustrated in Figure 2. The pipeline consists of two main stages: Subtask Synthesis and Task Synthe-

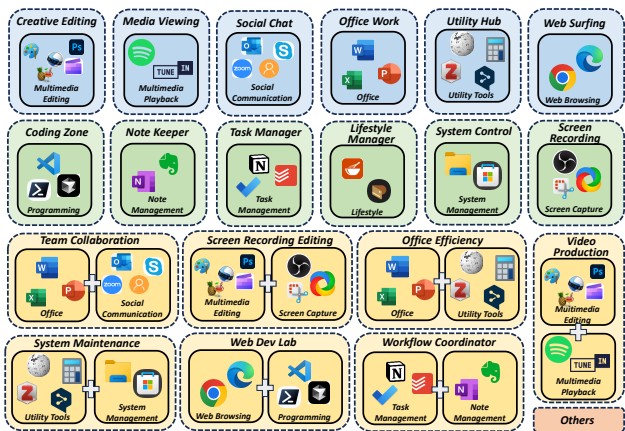

*Figure 9.* Overview of the 20 defined task scenarios. The scenarios are derived from combinations of application categories, where each box represents a task scenario with associated application icons.

sis. The Subtask Synthesis stage includes two steps: Subtask Discovery and Iterative Synthesis, while the Task Synthesis stage consists of Task Composition and Task Validation. In the following sections, we provide a detailed explanation of each component in this data synthesis pipeline.

### B.1. Subtask Discovery

To facilitate comprehensive subtask generation, we construct a dynamic environment that incorporates 49 heterogeneous applications, drawing inspiration from OSWorld (Xie et al., 2024). This environment enables MLLMs to interact with diverse functionalities, systematically analyze operational constraints, and propose well-structured subtasks tailored to each application's context. To support this process, we provide detailed API documentation, user manuals, and curated example subtasks, ensuring that MLLMs can infer practical usage scenarios. In addition, our approach emphasizes dependency modeling through a structured resource framework, where each subtask explicitly defines its required inputs and expected outputs. This predefined resource list serves as a guiding constraint, allowing MLLMs to reason about inter-subtask dependencies, avoid conflicts, and ensure smooth task execution. By leveraging this controlled exploration, MLLMs generate coherent and executable subtask sequences that align with real-world application workflows.

### B.2. Iterative Synthesis

**Trajectory Synthesis.** With the rapid development of MLLMs (Li et al., 2022; Pan et al., 2025b; 2024a; 2023), they are becoming increasingly capable (Li et al., 2020a; Pan et al., 2024c; 2025a;c). We deploy state-of-the-art MLLMs to execute the synthesized subtasks and record their exe-

cution traces as synthesized trajectories. Since subtasks are typically composed of a sequence of relatively simple instructions, advanced MLLMs achieve high success rates on these subtasks, making them suitable for generating an initial set of trajectories.

**Evaluation Synthesis.** Inspired by prior work (Gao et al., 2024b; 2025; Fei et al., 2024d;b; Li et al., 2023b; Miao et al., 2024), we first manually designed 11 system-level APIs, such as retrieving the text under the mouse cursor, accessing the clipboard content, and extracting visible text from the screen. We then provided the function signatures of these APIs to Code LLMs, which generated evaluation functions used to determine the completion status of sub-tasks. Since these evaluation functions are composed by invoking the designed APIs, we are able to assign more fine-grained evaluation scores based on the number of successfully executed API calls. Figure 10 illustrates the process of composing sub-task evaluation functions using these APIs.

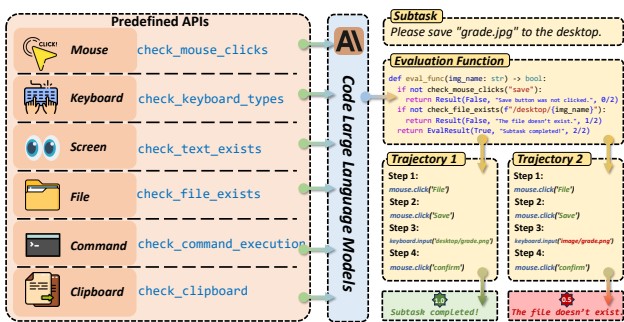

*Figure 10.* An overview of how predefined system-level APIs are composed into evaluation functions using Code LLMs.

**Cross-Verification.** We design a cross-verification algorithm to optimize synthesized subtasks. Specifically, the algorithm performs $N$ iterations, where an MLLM and a Code LLM specifically synthesize trajectories and evaluation functions based on the feedback from the previous iteration. The evaluation functions provide detailed failure feedback (*e.g.*, failing to click the "save" button), while the trajectories incorporate environmental state feedback, enabling an iterative refinement process for both trajectories and evaluation functions. To ensure the evaluation functions maintain sufficient discriminatory power, any function that passes cross-verification is further tested by evaluating three additional trajectories from other tasks.

### B.3. Task Composition

**Environmental resources for task composition.** To precisely capture the execution dependencies between different subtasks, we propose a set of concepts for environmental resources, as shown in Figure 11. Specifically, each environmental resource has a resource category and an actual

parameter. For example, we can use `img_path` to represent a category of images that exist locally, and the actual parameter `/usr/example.png` then instantiates this environmental resource. Each subtask has an input resource list and an output resource list, which represent the prerequisite resources required to execute the subtask and the new resources generated after execution, respectively. Through this method of resource representation and the logic of resource transformation, we can clearly define the dependencies between subtasks. A dependency relationship exists between two subtasks only if one subtask can provide the resources in the input resource list of the other subtask.

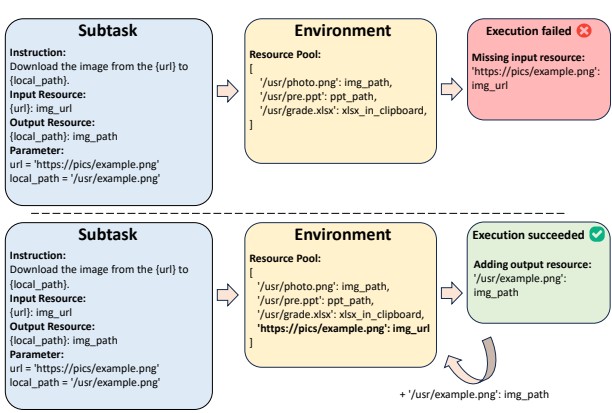

*Figure 11.* An overview of the constraint relationships among environmental resources.

**Task Composition with Consistent Intent.** To construct meaningful task graphs, we first curate a subtask pool by filtering out low-quality samples through a cross-verification process. A naïve bottom-up approach that directly connects subtasks based on shared input-output resources may generate incoherent tasks that lack a clear, unified goal. For example, an arbitrary combination might result in a task that simultaneously plays media from different applications without a meaningful relationship. To mitigate this issue, we explicitly extract overarching task intents from the subtask pool, such as 'create a personal introduction PowerPoint for Emily," as illustrated in Figure 1. Each intent acts as an anchor, grouping subtasks that contribute to the same high-level objective. The task graph is then constructed by linking these subtasks based on resource dependencies. Since the bottom-up composition follows predefined rules, we maintain strict control over the synthesis process, ensuring that the resulting task graphs exhibit logical coherence and adjustable complexity. This structured approach prevents the generation of ill-formed tasks while supporting diverse and meaningful compositions.

## B.4. Task Validation

To generate concise yet accurate task instructions for graph-structured tasks, we utilize GPT-4o to synthesize a task description by integrating subtask instructions and the structural information of the task graph. However, direct summarization may introduce inconsistencies, such as misrepresenting the graph's dependency structure or oversimplifying it into a linear sequence, thereby distorting the original task complexity. To address this issue, we designed a consistency validation mechanism to evaluate the fidelity of the summarized instruction. In this process, GPT-4o is prompted to infer subtask dependencies purely from the generated task instruction, without access to the original task graph. The inferred dependency structure is then compared against the ground-truth graph. If discrepancies arise—such as missing parallel execution paths or incorrect sequencing—the instruction is flagged for revision and must be re-summarized to better preserve the intended graph structure. This validation ensures that the final task instruction accurately reflects the hierarchical and branching nature of the original task graph, improving the clarity and correctness of the synthesized tasks.

## C. Details of OmniBench

In this section, we provide a detailed description of the data formats used in OmniBench. The data in OmniBench can be broadly categorized into five types: Subtask Metadata, Subtask Trajectory, Subtask Evaluation, Task Metadata, and Task Trajectory. For each category, we present the corresponding data schema along with representative examples. Finally, we provide a visualization example of a task graph.

### C.1. Subtask Metadata

Each subtask includes the following seven attributes:

- **id**: A UUID that uniquely identifies the subtask.

- **instruction_template**: A template of the subtask instruction containing parameter placeholders.

- **application**: The name of the application to which the subtask belongs.

- **available_parameters**: A list of all configurable parameter sets used to instantiate diverse instructions. Each element in the list represents one valid combination of parameters.

- **OS**: The operating system associated with the subtask.

- **input_resources**: A list of prerequisite resources required before executing the subtask.

- **output_resources**: A list of resources produced after the subtask is completed.

**Example of Subtask Metadata**

```json
{
    "id": "25e2a51e-c019-1a9a-0747-d↲
    6fe0e9d457d",
    "instruction_template": "Open
    '{xlsx_path}', select the all
    data, and copy it.",
    "application": "Excel",
    "available_parameters": [
        {
            "xlsx_path":
            "C:\\Users\\user\\Deskto↲
            p\\office\\The Evolution
            of Urbanization
            Rate.xlsx"
        }
    ],
    "OS": "Windows",
    "input_resources": [
        "xlsx_path"
    ],
    "output_resources": [
        "table_in_clipboard",
        "xlsx_in_processing"
    ]
}
```

## C.2. Subtask Trajectory

Each subtask trajectory includes the following five attributes:

- **trajectory_id**: A unique identifier for the trajectory.

- **instruction**: The instantiated instruction, generated by applying a specific parameter set from the subtask's `available_parameters`.

- **observations**: A list of screenshots representing each step in the trajectory.

- **actions**: A list of actions taken at each step of the trajectory.

- **subtask_id**: The identifier of the subtask to which this trajectory corresponds.

**Example of Subtask Trajectory**

```json
{
    "trajectory_id": "XXX",
    "instruction": "Using the file
    explorer, navigate to C:\\Users\↲
    \user\\Desktop\\images\\ and new
    a Text Document named
    introduction.txt",
    "observations": [
        "obs1.png", "obs2.png", "..."
    ],
```

```json
    "actions": [
        {
            "function":
            "click_input",
            "args": {
                "button": "left",
                "double": false
            },
            "rect": [
                124,
                1020,
                179,
                1080
            ],
            "description": "There are
            many application icons on
            the taskbar, and I need
            to select the File
            Explorer to complete the
            task.",
            "thought": "To fulfill
            'Using the file explorer,
            navigate to C:\\Users\\u↲
            ser\\Desktop\\images\\
            and new a Text Document
            named introduction.txt',
            I need to first click the
            'File Explorer' button to
            open the corresponding
            application.",
            "control_text": "File
            Explorer"
        },
        {
            "function": "...",
            "args": { "...": "..." }
        }
    ],
    "subtask_id": "XXX"
}
```

## C.3. Subtask Evaluation

Each subtask evaluation is implemented as a Python function that checks for a sequence of expected interaction outcomes based on predefined system-level APIs. The function returns an `EvalResult` object containing three fields:

- **success**: A boolean value indicating whether the subtask was completed successfully.

- **message**: A string providing a human-readable explanation of the evaluation result.

- **progress**: A float or fraction representing the proportion of evaluation conditions that were satisfied, enabling partial credit.

Each evaluation function typically performs multiple checks, such as verifying whether specific UI elements were

clicked, expected text appeared on the screen, or input was typed correctly. These checks are implemented using task-agnostic API functions (e.g., `check_mouse_clicks`, `check_text_exists_via_control`, `check_keyboard_types`). By combining these low-level signals, the evaluation function provides a fine-grained and automated judgment of subtask execution.

---

**Example of Subtask Evaluation**

```python
from collections import namedtuple

EvalResult = namedtuple('EvalResult',
['success', 'message', 'progress'])

def evaluate_agent_task_completion(c
sv_path: str) ->
EvalResult:
    if not
    check_mouse_clicks(text='More
    actions'):
        return EvalResult(False,
        "Subtask execution failed
        because agent did not click
        the 'More actions' button.",
        0/4)

    if not check_text_exists_via_con
    trol(text='Import tasks from a
    spreadsheet using a CSV file.'):
        return EvalResult(False,
        "Subtask execution failed
        because the import tasks
        option was not accessed.",
        1/4)

    if not check_keyboard_types(text
    =csv_path):
        return EvalResult(False,
        f"Subtask execution failed
        because the CSV file path
        '{csv_path}' was not typed.",
        2/4)

    if not
    check_mouse_clicks(text='Open'):
        return EvalResult(False,
        "Subtask execution failed
        because the 'Open' button was
        not clicked to import the
        file.", 3/4)

    return EvalResult(True, "Subtask
    completed successfully", 4/4)
```

---

## C.4. Task Metadata

Each task includes the following four attributes:

- **task_instruction**: The natural language instruction that describes the overall task.

- **dag**: A directed acyclic graph (DAG) representing the structural dependencies among subtasks within the task.

- **task_intent**: The high-level goal or intent that the task is designed to achieve.

- **successful_topo**: All valid topological orders of the DAG that lead to successful task completion.

---

**Example of Task Metadata**

```json
{
    "task_instruction": "In Excel,
    open 'C:\\Users\\user\\Desktop\\
    office\\The Evolution of
    Urbanization Rate.xlsx', select
    the 'A' column, and center the
    content. Then, export the
    document as a PDF named 'C:\\Use
    rs\\user\\Desktop\\pdf\\The
    Evolution of Urbanization
    Rate.pdf'.",
    "dag": {
        "nodes": [
            "a7310aa0-b194-77e3-5c36
            -996391a1bc7d",
            "df3fc68b-fa76-4e19-7da6
            -aef17792523b"
        ],
        "edges": {
            "a7310aa0-b194-77e3-5c36
            -996391a1bc7d":
            [
                "df3fc68b-fa76-4e19-
                7da6-aef17792523b"
            ],
            "df3fc68b-fa76-4e19-7da6
            -aef17792523b":
            []
        }
    },
    "task_intent": "Center Excel data
    and export to PDF",
    "successful_topo": [
        [
            "a7310aa0-b194-77e3-5c36
            -996391a1bc7d",
            "df3fc68b-fa76-4e19-7da6
            -aef17792523b"
        ]
    ]
}
```

---

## C.5. Task Trajectory

Each task trajectory includes the following seven attributes:

- **trajectory_id**: The unique identifier of the trajectory. The suffix (0) indicates that it corresponds to the first topological order in the `successful_topo` list.

- **task_id**: The identifier of the associated task.

- **topological_order**: The specific topological order followed in this trajectory.

- **instruction**: The instruction describing the overall task.

- **intent**: The high-level intent or goal of the task.

- **observations**: The sequence of visual observations recorded during task execution.

- **actions**: The sequence of actions taken during task execution.

---

**Example of Task Trajectory**

```
{
  "trajectory_id": "12(0)",
  "task_id": "12",
  "topological_order": [
    "a7310aa0-b194-77e3-5c36-996391a
    1bc7d",
    "df3fc68b-fa76-4e19-7da6-aef1779
    2523b"
  ],
  "instruction": "In Excel, open the
  file, center the A column, and
  export as PDF.",
  "intent": "Center Excel data and
  export to PDF",
  "observations": [
    "obs1.png", "obs2.png", "..."
  ],
  "actions": [
    {
      "function": "click_input",
      "args": {
        "button": "left",
        "double": true
      },
      "rect": [1520, 371, 1614, 458],
      "description": "Double-click
      the 'Excel' icon on the
      desktop.",
      "thought": "To begin the task,
      I need to open Excel.",
      "control_text": "Excel"
    },
    {
      "function": "...",
      "args": { "...": "..." }
    }
  ]
}
```

---

### C.6. Example of Task Graph

To more clearly illustrate the relationships between subtasks and tasks, we provide an additional visualization example of

a task graph, as shown in Figure 12. Each node in the graph signifies a distinct subtask, categorized by the application used: PowerPoint (orange), Outlook (blue), and Photoshop Express (yellow). The directed edges denote the sequence of execution, where each subtask must be completed before the subsequent ones can begin.

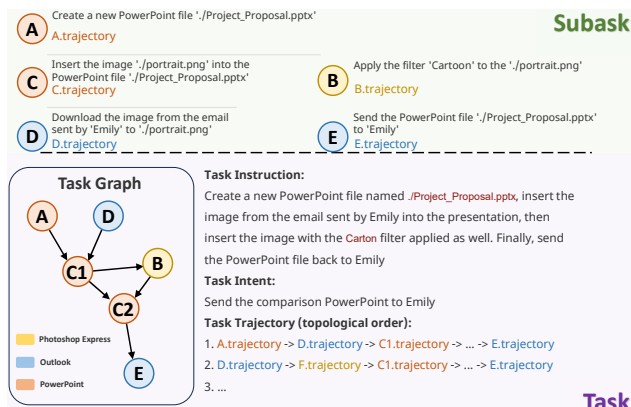

*Figure 12.* A task example in OmniBench.

## D. Details of OmniEval

In this section, we introduce the details of OmniEval, including the design of ten essential capabilities, the experimental setup for evaluation, and implementation details during the evaluation.

### D.1. Capability Design

As shown in Table 12, we construct 10 specialized sets of test tasks based on the five-dimensional complexity, each set designed to evaluate one of the 10 capabilities required for agents to fulfill user requests. The number of solid stars represents the level of complexity, while zero stars indicate that the complexity in that dimension can be arbitrary.

| | Dependency | Instruction | Hierarchy | Branch | Knowledge |
|---|---|---|---|---|---|
| Parallel Planning | ★★★ | ☆☆☆ | ☆☆☆ | ★★★ | ☆☆☆ |
| Long-Range Planning | ★★★ | ☆☆☆ | ★★★ | ☆☆☆ | ☆☆☆ |
| Long-Sequence Reasoning | ☆☆☆ | ★★★ | ★★★ | ☆☆☆ | ☆☆☆ |
| Long Instruction Following | ☆☆☆ | ☆☆☆ | ★☆☆ | ★★★ | ☆☆☆ |
| Sequential Decision-Making | ☆☆☆ | ☆☆☆ | ★★★ | ★★★ | ☆☆☆ |
| Cross-Domain Decision-Making | ☆☆☆ | ☆☆☆ | ☆☆☆ | ★★★ | ★★★ |
| Subtask Identification | ★☆☆ | ★★★ | ☆☆☆ | ☆☆☆ | ☆☆☆ |
| Dependency Identification | ★★★ | ★☆☆ | ☆☆☆ | ☆☆☆ | ☆☆☆ |
| Cross-Domain Knowledge | ☆☆☆ | ★★★ | ☆☆☆ | ☆☆☆ | ★★★ |
| Domain-Specific Knowledge | ☆☆☆ | ★★★ | ☆☆☆ | ☆☆☆ | ★☆☆ |

*Table 12.* Constraining five fundamental task complexities (top) to construct test tasks across ten capability dimensions (left).

### D.2. Experiment Setup

#### D.2.1. SETTINGS

For the evaluation phase, we follow the practices of works such as Aguvis and UGround, ensuring consistency in pre-

processing and maintaining comparability across different models. Specifically, we standardize the image resolution by scaling all input images to $1024 \times 1024$ pixels. This resolution is chosen to balance computational efficiency and visual detail, ensuring that models can effectively process graphical user interfaces (GUIs) without excessive memory overhead or loss of important information. The decision to use a fixed resolution is motivated by several key factors. First, many modern multimodal models and virtual agents, including those designed for GUI interaction, are trained on datasets with varying resolutions. Standardizing the input resolution reduces inconsistencies that may arise due to different input scales, helping models generalize better across diverse GUI layouts. Additionally, this resolution aligns with commonly used image sizes in recent benchmarks, facilitating direct comparison with prior works. Moreover, scaling images to $1024 \times 1024$ ensures that finer details within the GUI, such as text labels, buttons, and icons, remain distinguishable without introducing excessive noise or aliasing effects. Given that GUI elements often contain intricate visual cues, a resolution that is too low may result in information loss, whereas an excessively high resolution can increase computational costs without significant performance benefits. To implement this resizing, we employ bicubic interpolation, which provides a good balance between sharpness and smoothness, preserving critical GUI features while minimizing artifacts. We also ensure that aspect ratios are maintained whenever possible by applying padding techniques if necessary, preventing unintended distortions that could affect model predictions. By adopting this resolution standard, we align our evaluation methodology with established works like Aguvis and UGround, fostering reproducibility and enabling fair comparisons in GUI-based task assessments.

### D.2.2. BASELINES

We conducted a comprehensive evaluation of the following four types of models:

**1) Closed-source MLLMs**: These include GPT-4o (Hurst et al., 2024), Qwen-VL-Max (Bai et al., 2023), Claude-3.5-Sonnet, and Gemini-2.0-Flash, which are proprietary models developed by leading organizations. These models are selected due to their state-of-the-art performance on a wide range of multimodal tasks, including vision-language understanding, reasoning, and instruction following. Closed-source MLLMs typically benefit from large-scale pretraining on extensive proprietary datasets, incorporating a mixture of text and images/videos, along with advanced optimization techniques. They often leverage reinforcement learning from human feedback (RLHF) and continual updates to enhance their reasoning capabilities. These models are accessible via APIs, which enforce constraints on inference settings such as token limits, response latency, and

proprietary decoding strategies. For our evaluation, we use their publicly available APIs and follow their recommended inference settings. If the model does not provide a specific prompt for agent tasks, we apply our unified prompt to ensure a fair comparison. Additionally, since these models lack direct access to GUI-specific training data, we evaluate their adaptability by providing structured information about the interface components.

**2) Open-source MLLMs**: We also evaluate open-source models such as Qwen2-VL-7B-Instruct (Wang et al., 2024b), InternVL2-8B (Chen et al., 2024), and InternVL2.5-8B (Chen et al., 2024), which are widely recognized in the research community for their flexibility and strong performance. These models have been fine-tuned on large-scale multimodal datasets and provide strong generalization across diverse vision-language tasks. Open-source MLLMs offer several advantages, including transparency in training methodologies, customizability for domain-specific fine-tuning, and community-driven improvements. Unlike closed-source models, researchers can inspect their architectures, modify their training pipelines, and deploy them on local hardware without API restrictions. We use their fine-tuned weights and apply our unified prompt for tasks where no default prompt is provided. Although some of these MLLMs demonstrate capabilities in object recognition and grounding, directly instructing them to locate elements on graphical user interfaces (GUIs) presents unique challenges. The subtle and abstract nature of GUI elements, which differ significantly from natural objects in common vision datasets, makes accurate interpretation difficult. To mitigate this issue, we provide A11Y (accessibility) metadata, such as element descriptions and structural information, to aid inference.

**3) Virtual Agents**: In addition to MLLMs, we also evaluate various virtual agents, such as Aguvis (Xu et al., 2024b), OS-Atlas (Wu et al., 2024b), and ShowUI (Lin et al., 2024). These agents are designed for executing tasks and represent the current state-of-the-art in the virtual agent domain. Unlike MLLMs, which primarily operate as general-purpose multimodal models, virtual agents are often specialized for task execution. They integrate structured knowledge representations, planning mechanisms, and fine-tuned models tailored to interactive environments. Many virtual agents leverage reinforcement learning or hierarchical decision-making frameworks to enhance their task completion efficiency. Some also incorporate retrieval-based techniques to access domain-specific knowledge dynamically. Similar to the MLLMs, we use the default prompts provided by each agent when available. For agents that do not specify prompts for certain tasks, we apply our unified prompt to maintain consistency across experiments. Given that virtual agents typically rely on structured inputs rather than free-form multimodal understanding, we evaluate their ability

to process GUI environments by simulating real-world task execution scenarios.

**4) Supervised Fine-Tuning Agents**: We selected two backbones, OS-Atlas-Base-4B (Wu et al., 2024b) and UGround-V1-7B (Gou et al., 2024), with different architectures and fine-tuned them using the synthetic data from OmniBench. Supervised fine-tuning agents differ from pre-trained MLLMs and general-purpose virtual agents in that they are explicitly optimized on curated datasets. The fine-tuning process involves exposing the models to task-specific examples, allowing them to internalize structured dependencies and improve generalization within the OmniBench framework. This approach enables agents to learn robust action sequences, refine their perception of GUI elements, and enhance their decision-making accuracy. By leveraging synthetic data, we ensure that these agents develop a structured understanding of GUI tasks while minimizing biases inherent in real-world datasets. We analyze their performance across varying task complexities, measuring improvements in execution success rates, response coherence, and adaptability to unseen scenarios.

### D.3. Evaluation Details

In this section, we outline the evaluation strategies adopted for different categories of models, considering their architectural characteristics and observed behaviors.

For open-source MLLMs, we observe that models such as Qwen-VL exhibit limited ability to process high-resolution content across multiple images, often resulting in severe hallucinations. To mitigate this, we simplify the visual input by providing only a single high-resolution screenshot of the current state. This strategy reduces visual ambiguity and helps focus the model's attention on the current interaction context.

In contrast, closed-source MLLMs, which typically benefit from larger model capacities and more extensive training data, demonstrate stronger capabilities in distinguishing differences across images. For these models, we provide two concatenated images: (1) the screenshot of the previous click location with red box annotations, and (2) the current high-resolution screenshot with similar markup. These two images are horizontally stitched into a single input to highlight temporal context and action history.

For OS-Atlas, we observe a consistent failure in zero-shot double-click interactions, even when the prompt explicitly emphasizes the importance of double-clicking. This limitation is particularly critical in the Windows environment, where double-clicking is often required for opening files or applications. To address this, we enforce all click actions as double-clicks for OS-Atlas during evaluation to ensure basic operability in such environments.

## E. Prompts

In this section, we present the prompts used for constructing subtask trajectories and subtask evaluations.

### E.1. Prompt for Subtask Trajectory Synthesis

**Example Prompt**

```
system: |-
  You are now operating in Executable
  Language Grounding mode. Your task
  is to help users accomplish their
  goals by suggesting executable
  actions based on the provided task
  instructions and your observations
  of the current situation.

  ## Environment Interaction Rules

  ### Screenshots
  - You are provided two versions of
  screenshots of the current
  application in a single image, one
  with annotation (right) and one
  without annotation (left)
  - The annotation is to help you
  identify the control elements on
  the application
  - The annotation is a small
  rectangle with a number in the
  center of the rectangle in the top
  left corner of the control item.
  The number is the label of the
  control item
  - Different types of control items
  have different colors of annotation
  ### Control Items
  - The control item is the element
  on the page that you can interact
  with, such as button, input box,
  etc.
  - You are given the information of
  all available control items in the
  current application window in a
  list format:
    {{
      "label": <The annotated label
      of the control item>,
      "control_text": <The text of
      the control item>,
      "control_type": <The type of
      the control item>,
      "parent_control_text": <The
      text of the parent control
      item. When you are not sure
      which control to select, you
      can make a decision based on
      their parent controls>,
```

```
        "parent_control_type": <The
        type of the parent control
        item. When you are not sure
        which control to select, you
        can make a decision based on
        their parent controls>
    }}
### Control Operations
- You are able to use pywinauto to
interact with the control item
{apis}

### Execution Status
- You are required to determine the
status of the execution after
performing the current action.
Choose from the following options
and fill in the "Status" field in
the response:
    - "CONTINUE": means the task is
    not yet completed and further
    actions are required. This is
    typically chosen when the
    execution is still ongoing or
    needs additional steps.
    - "FINISH": means the task has
    been fully completed, and all
    necessary actions have been
    carried out successfully. Only
    choose this when all steps have
    been executed as planned and the
    task is considered finished.

### Other Guidelines
- You are required to respond in a
JSON format, consisting of 8
distinct parts with the following
keys and corresponding content:
    {{
        "Status": <Specify the status
        of the exploration. If "Status"
        is "FINISH", the
        "ControlLabel", "ControlText",
        "Function", and "Args" should
        be empty>,
        "Observation": <summarize the
        screenshot from the previous
        step, if it exists.  You can
        also compare the current
        screenshot with the one taken
        at the previous step>,
        "Thought": <Outline your
        thinking and logic of the
        current one-step action
        required to seek inspiration
        for task design>,
```

```
        "ControlLabel": <Specify the
        precise annotated label of the
        control item to be selected,
        adhering strictly to the
        provided options in the field
        of "label" in the control
        information. If you believe
        none of the control items are
        suitable for the task or the
        task is complete, kindly output
        an empty string ''>,
        "ControlText": <Specify the
        precise control_text of the
        control item to be selected,
        adhering strictly to the
        provided options in the field
        of "control_text" in the
        control information. If you
        believe none of the control
        items are suitable for the task
        or the task is complete, kindly
        output an empty string ''. The
        control text must match exactly
        with the selected control
        label>,
        "Function": <Specify the
        precise API function name
        without arguments to be called
        on the control item to complete
        the user request, e.g.,
        click_input. Leave it an empty
        string "" if you believe none
        of the API functions are
        suitable for the task or the
        task is complete>,
        "Args": <Specify the precise
        arguments in a JSON object
        format of the selected API
        function to be called on the
        control item to complete the
        user request, e.g., {{"button":
        "left", "double": false}}.
        Leave it an empty dictionary
        {{}} if the API does not
        require arguments, or you
        believe none of the API
        functions are suitable for the
        task, or the task is complete>,
    }}

  Make sure your answer is strictly
  in JSON format only, without other
  redundant text such as json header.
  Your output must be able to be
  parsed by json.loads(). Otherwise,
  it will crash the system and
  destroy the user's computer.

user: |-
  <Step History:> {action_history}
  <Available Control Item:>
  {control_item}
```

```
<Task instruction:>
{task_instruction}
```

## E.2. Prompt for Subtask Evaluation Synthesis

**Example Prompt**

```
system: |-
  You are a coding assistant tasked
  with generating Python code to
  evaluate if a digital agent has
  successfully completed a specific
  task. You will receive a task
  description along with a set of
  APIs that you can use to check
  different actions or conditions
  that indicate task completion. Your
  goal is to write an evaluation
  function that returns True if the
  agent has successfully completed
  the task and False otherwise.

  ### Available APIs:
  ```python
  def check_mouse_clicks(text: str)
  -> bool:
    """Checks if the mouse has
    clicked on the specified text.
    Parameters
    ---------
    text: str
        The text associated with the
        click.
    Returns
    ---------
    bool
        True if the mouse has clicked
        on the specified text, False
        otherwise.
    Examples
    ---------
    >>> # Evaluate if the agent has
    successfully set the picture
    'envelope.png' as background
    >>> def
    evaluate_agent_task_completion():
    >>>     if not check_mouse_click⌋
    s(text='envelope.png'):
    >>>         return False
    >>>     if not
    check_mouse_clicks(text='set as
    background'):
    >>>         return False
    >>>     return True
    """

  def check_keyboard_types(text: str)
  -> bool:
    """Checks if the keyboard has
    typed the specified text.
```

```
    Parameters
    ---------
    text: str
        The text to be typed.
    Returns
    ---------
    bool
        True if the keyboard has
        typed the specified text,
        False otherwise.
    Examples
    ---------
    >>> # Evaluate if the agent has
    successfully typed 'Hello,
    World!'
    >>> def
    evaluate_agent_task_completion():
    >>>     if not
    check_keyboard_types(text='Hello,
    World!'):
    >>>         return False
    >>>     return True
    """

  def check_file_exists(file_path:
  str) -> bool:
    """Checks if the specified file
    exists.
    Parameters
    ---------
    file_path: str
        The path to the file to be
        checked.
    Returns
    ---------
    bool
        True if the file exists,
        False otherwise.
    Examples
    ---------
    >>> # Evaluate if the agent has
    successfully renamed 'cat.jpg' to
    'cute cat.jpg'
    >>> def
    evaluate_agent_task_completion():
    >>>     if check_file_exists(fil⌋
    e_path='C:/Users/user/Desktop/im⌋
    ages/cat.jpg'):
    >>>         return False
    >>>     if not
    check_file_exists(file_path='C:/⌋
    Users/user/Desktop/images/cute
    cat.jpg'):
    >>>         return False
    >>>     return True
    """

  def check_text_exists_via_ocr(text:
  str) -> bool:
    """Checks if the specified text
    is present in the last screenshot
    using OCR (Optical Character
    Recognition).
```

```
    Parameters
    ---------
    text: str
        The text to be checked.
    Returns
    ---------
    bool
        True if the text is present
        in the last screenshot, False
        otherwise.
    Examples
    ---------
    >>> # Evaluate if the agent has
    successfully set the clock to
    '9:00 AM'
    >>> def
    evaluate_agent_task_completion():
    >>>     if not check_text_exists⌋
    _via_ocr(text='9:00
    AM'):
    >>>         return False
    >>>     return True
    """

def
check_text_exists_via_control(text:
str) -> bool:
    """Checks if the specified text
    is present in the last screenshot
    through control information.
    Parameters
    ---------
    text: str
        The text to be checked.
    Returns
    ---------
    bool
        True if the text is present
        in the last screenshot, False
        otherwise.
    Examples
    ---------
    >>> # Evaluate if the agent has
    successfully input the code
    'print("Hello World!")'
    >>> def
    evaluate_agent_task_completion():
    >>>     if not check_text_exists⌋
    _via_control(text='print("Hello
    World!")'):
    >>>         return False
    >>>     return True
    """

def check_text_exists(text: str) ->
bool:
    """Checks if the specified text
    is included in the last
    screenshot.
    Parameters
    ---------
    text: str
        The text to be checked.
```

```
    Returns
    ---------
    bool
        True if the text is present
        in the last screenshot, False
        otherwise.
    Examples
    ---------
    >>> # Evaluate if the agent has
    successfully created a new folder
    named 'Project Files'
    >>> def
    evaluate_agent_task_completion():
    >>>     if not
    check_text_exists(text='Project
    Files'):
    >>>         return False
    >>>     return True
    """
```

### Other Guidelines
- You will be given a `Subtask Instruction Template` and `Parameters`. Use the APIs provided to implement an `Evaluation Function` in Python.
- This agent will run on the `Windows 11` operating system, so please consider how to cleverly design the evaluation function based on this operating system.
- The evaluation function should return a namedtuple `EvalResult` with two fields:
    - `success`: A boolean indicating if all conditions are met (True) or not (False)
    - `message`: A string explaining why the evaluation succeeded or failed
- The evaluation function should check each required condition and return appropriate success/failure messages.
- Please `directly output` the evaluation function, without any additional comments or explanations.
- When you design a correct evaluation function, I will provide you with a `$1000` tip.

**user**: |-
  ### Subtask Instruction Template
  {instruction}

  ### Available Parameters
  {parameters}

  ### Controls in Environment
  {controls}

