# OpenReview forum: "What Limits Virtual Agent Application? OmniBench: A Scalable Multi-Dimensional Benchmark for Essential Virtual Agent Capabilities"
_ICML.cc/2025/Conference — ICML 2025 oral_

### Official Review · Reviewer_7v9F · 2025-03-08

**Overall Recommendation:** 4

**Summary:**

This submission introduces OmniBench, aiming to provide a scalable task synthesis paradigm along with an evaluation framework for agent capabilities across ten dimensions. Due to the inherent complexity of agent trajectories, it is challenging to effectively construct large-scale, high-quality trajectory datasets. Existing trajectory datasets in agent community are typically three orders of magnitude smaller than GUI grounding datasets (thousands vs. millions), struggling to provide sufficient supervision to enhance agents' navigation capabilities across various environments.

To construct tasks more systematically, the authors propose the Task Graph, a novel task modeling approach. They treat frequently occurring subtasks as nodes in the task graph. First, they synthesize subtasks and apply cross-verification to obtain gold subtasks. Then, they synthesize tasks through rule-based subtask composition. This synthesis pipeline not only avoids the need for strong task planning capabilities in top-down task decomposition but also exponentially scales up task synthesis by reusing synthetic gold subtasks.

Additionally, the authors quantifies task complexity based on five fundamental attributes of task graph (e.g., depth, number of nodes, number of edges, etc.). By constraining the complexity levels of the five dimensions, the authors filter test tasks for evaluating 10 capabilities and reveal the fine-grained performance of 12 agents, providing guidance to the community on the direction of future optimization.

## update after rebuttal
Authors' rebuttal have solved my concerns. I think previous score is high enough and I will keep my rating.

**Claims And Evidence:**

Yes, the claims made in the submission are supported by clear and convincing evidence. I have reviewed the submittion and did not find any inappropriate claims.

**Essential References Not Discussed:**

I believe the inspiration for evaluating various agent capabilities in the submission might come from benchmarks [1, 2] used to evaluate MLLM capabilities. However, this is not explicitly discussed in the submission. Including such a discussion would make the submission more impactful.

[1] MMBench: Is Your Multi-modal Model an All-around Player?

[2] SEED-Bench: Benchmarking Multimodal LLMs with Generative Comprehension

**Experimental Designs Or Analyses:**

Yes, I have checked the experimental designs and analyses. As a benchmark, I consider the experiments to be sound and valid. If the authors can further validate the quality of the training data, this submission would have a greater impact.

**Methods And Evaluation Criteria:**

Yes, I think the methods and evaluation criteria make sense. Considering agent tasks as a directed acyclic graph provides a rational foundation for subsequent task synthesis, task complexity definition, and agent capability design. Additionally, Figure 6 shows that OmniEval aligns with human evaluation, demonstrating that the evaluation criteria also make sense.

**Other Comments Or Suggestions:**

1. The metric ND under "Overall" in Table 4 has not appeared before. I suspect this is a typo by the authors.

2. Change 'we introduced OmniBench' in line 429 on the right side, 'we observed a marked' in line 431 on the left side, and 'We compared' in line 378 on the left side to the present simple tense.

3. The line spacing around line 266 seems too tight and should be adjusted.

4. In the subfigure at the bottom left of Figure 1, the authors may have forgotten to replace 'x' with the actual values. Additionally, this figure does not seem to intuitively illustrate that task complexity has three levels and its relationship with agent capabilities.

**Other Strengths And Weaknesses:**

## Strengths

S1. The authors propose a data collection pipeline to automatically synthesize high-quality demonstration trajectories and evaluation functions. The author designed three quality control modules to ensure the data quality, achieving a 91% human acceptance rate. The proposed bottom-up synthesis provides a highly valuable insight, which can be important for the community: fully leveraging the repeatedly occurring sub-trajectory in trajectory data.

S2. The authors introduce the concept of the Task Graph, allowing for a systematic quantification of task complexity based on fundamental graph properties (e.g., depth, number of edges, etc.). By constraining the complexity levels of the five dimensions, the authors filter test tasks for evaluating 10 capabilities.

S3. Experiments are conducted to evaluate agents across 10 capability dimensions, revealing the fine-grained performance differences between open-source and closed-source agents, providing precise directions for future optimization.

## Weaknesses

W1. On the left side of lines 430 to 434, the authors state that "By incorporating intents into the prompt, we observed a marked improvement in the agents’ performance on tasks designed to evaluate planning." Additionally, Figure 9 illustrates the impact of intent on OS-Atlas-Base and Uground-V1 in the agent setting, showing a significant improvement in planning capabilities when task intent is incorporated.

However, the intent here is actually provided as input to the planner (i.e., GPT-4o), and the authors don’t test the effect of feeding intent into end-to-end agents. If task intent is only effective on GPT-4o, it is hard to prove that task intent is as meaningful as stated in the submission.

W2. The subfigure in the bottom left of Figure 1 shows that the graph-based data is of high quality, but I couldn't find any related experimental descriptions in the article. I believe that comparing the synthetic trajectories with human demonstrations could reveal the quality of the OmniBench dataset. Therefore, I suggest that the authors train the agent separately on these two types of data and evaluate its performance on other benchmarks.

W3. The authors define task complexity from five dimensions in Table 2 but don’t evaluate the performance of agents on tasks within a specific dimension. The experimental results are valuable to the community, as they could reveal whether certain models exhibit preferences for specific types of tasks.

**Questions For Authors:**

Q1. How is the "predefined resource list for determining subtask inputs and outputs" mentioned in the submission implemented? Does the predefined resource list come from the LLM? What are the design principles for the resources?

Q2. I noticed that the appendix of OmniBench introduces several apps in the environment, including some that require an internet connection (e.g., Skype, Mail, DeepL). How do the authors ensure that the environment remains consistent for each evaluation? I am curious about this because interactive benchmarks [1, 2, 3] typically include only offline apps/websites (e.g., Microsoft Office, VS Code) to ensure a consistent environment for each evaluation.

[1] AndroidWorld: A Dynamic Benchmarking Environment for Autonomous Agents

[2] OSWorld: Benchmarking Multimodal Agents for Open-Ended Tasks in Real Computer Environments

[3] WebArena: A Realistic Web Environment for Building Autonomous Agents

**Relation To Broader Scientific Literature:**

OmniBench provides a more fine-grained evaluation of agents, unlike most benchmarks that focus on task success rates. It evaluates ten diverse capabilities, including dependency identification and long-range planning, similar to benchmarks like MMBench designed for MLLMs. However, in the agent domain, this might be the first attempt I am aware of to define the fine-grained capabilities required for agents.

**Theoretical Claims:**

I have reviewed the proofs for the theoretical claims and consider them to be correct.

---

> ### Author Rebuttal · Authors · 2025-04-01
>
> Thank you for appreciating our paper as comprehensive and identifing highly valuable insight, your constructive comments and suggestions are valuable to us. Below is our detailed response to clarify the points you raised.
>
> **Q1: The Effect of Task Intent on Planning.**
>
> **A1:** The task intent is to enhance the planning capability of the planner in a plug-and-play way. We further explore the generalizability of task intents on the **open-source and closed-source models**. (1) For open-source models, we fine-tune OS-Atlas-Base-4B and UGround-V1-7B in the same dataset with and without task intent, respectively. As shown in the following table, incorporating task intent in the training data significantly improves the model's planning performance on OmniBench. This indicates that task intent has the potential to serve as fine-tuning data to guide models in improving their planning capabilities.
> |Models| Parallel Planning|Long-Range Planning|
> |-|-|-|
> |Omni-OS-Atlas-Base-4B|24.2|33.0|
> |**+intent tuning**|**25.7 (+1.5)**|**34.9 (+1.9)**|
> |Omni-UGround-V1-7B| 33.2|31.3|
> |**+intent tuning**|**34.4 (+1.2)**|**32.6 (+1.3)**|
>
> (2) For closed-source models, we use prompts with and without task intent to Qwen-VL-Max, Gemini-2.0-Flash, and Claude-3.5-Sonnet for planners, with UGround-V1-7B serving as the grounding models.
>
> |Models|Parallel Planning|Long-Range Planning|
> |-|-|-|
> |Qwen-VL-Max|21.9|20.8|
> |**+intent prompt**|**24.5 (+2.6)**|**23.5 (+2.7)**|
> |Gemini-2.0-Flash|23.1|22.7|
> |**+intent prompt**|**28.9 (+5.8)**|**26.7 (+4.0)**|
> |Claude-3.5-Sonnet| 24.2|23.7|
> |**+intent prompt**|**30.6 (+6.4)**|**28.1 (+4.4)**|
>
> &nbsp;
>
> **Q2: Compare with Human Demonstration Trajectories.**
>
> **A2:** Thank you for nicely pointing out that Omnibench may obtain high-quality graph-based tasks. To address this, we compare human demonstration trajectories from the GUIAct dataset and synthetic trajectories from OmniBench. We conduct experiments on the OmniAct-Web dataset and apply the same fine-tuning settings for fair comparison.
>
> |Models|Type-Web|Grounding-Web|SR-Web|
> |-|-|-|-|
> |InternVL2-4B|47.51|51.34|24.39|
> |Qwen2-VL-7B|89.22|85.94|78.58|
> |SeeClick|86.98|75.48|68.59|
> |OS-Atlas-4B|88.56|82.00|73.91|
> |&nbsp; +1k human demostrations|88.64|82.34|74.06|
> |&nbsp; +1k synthesized trajectories|**88.71**|**82.50**|**74.12**|
> |UGround-7B-V1|90.16|86.98|79.85|
> |&nbsp; +1k human demostrations|90.23|87.19|80.02|
> |&nbsp; +1k synthesized trajectories|**90.29**|**87.28**|**80.11**|
>
> The results in above table show that **graph-based trajectories in OmniBench are of high quality and lead to better performance**.
>
> &nbsp;
>
> **Q3: Performance per Complexity Dimension.**
>
> **A3:** Thanks for your constructive suggestions. We report the model's performance across different levels within each complexity dimension in the table.
>
> |Models|D-Easy|D-Medium|D-Hard|B-Easy|B-Medium|B-Hard|I-Easy|I-Medium|I-Hard|K-Easy|K-Medium|K-Hard|H-Easy|H-Medium|H-Hard|
> |-|-|-|-|-|-|-|-|-|-|-|-|-|-|-|-|
> |Aguvis-7B|32.8|27.6|24.3|41.2|36.8|30.6|49.5|36.9|25.3|38.4|32.5|27.6|37.9|33.6|29.7|
> |OS-Atlas-Pro-7B|32.3|26.8|23.7|39.1|31.0|25.4|44.3|34.8|21.8|33.9|28.4|24.3|34.5|28.1|25.6|
> |ShowUI-2B*|34.0|28.3|25.6|41.3|32.7|28.2|45.9|36.6|25.4|37.8|32.6|27.4|37.6|32.0|28.1|
> |OS-Atlas-Base-4B*|32.7|29.1|24.9|35.2|32.4|27.6|48.2|37.5|26.7|39.1|34.2|28.9|43.1|38.4|33.2|
> |UGround-7B*|34.1|30.0|27.1|44.6|38.3|32.4|53.0|39.4|27.2|42.3|36.4|32.6|35.7|28.8|25.5|
>
> where D means Dependency, B means Branching, I means Instruction, K means Knowledge, and H means Hierarchical. As can be seen, the model's performance gradually decreases as the complexity increases, indicating that the **task complexity is reasonable and independent**.
>
> &nbsp;
>
> **Q4: How to design predefined resource lists for determining subtask inputs and outputs?**
>
> **A4:** Thanks for your interest in the details of our subtask determination. To capture the specific state of the virtual environment, we manually design cold start resources based on the principles of **subtask definition and task transition** to ensure accuracy and facilitate seamless integration between subtasks. To further explain the implementation, we provide additional examples of input and output resources in Figure 1 via the anonymous link: https://anonymous.4open.science/r/OmniBench_rebuttal-A4C7/r4.md
>
> **Q5: How to ensure environmental consistency?**
>
> **A5:** Thanks for your comment, we also believe this verification is important. Since there is an inherent trade-off between environmental realism and consistency, when introducing web-based applications~(e.g., Skype, Mail, DeepL) to the OmniBench evaluation environment, we will **use firewalls to block their outbound traffic and preload the necessary page state** to provide an environment that ensures realism and consistency.
>
> **Q6: Writing issues.**
>
> **A6:** We sincerely thank you for constructive suggestions on expression, organization, figures, and citations. We will correct them in next version.

---

> > ### Comment · Reviewer_7v9F · 2025-04-02
> >
> > Thanks for the author's response. I believe it has addressed all my concerns.
> > Once again, I find the ideas meaningful and thought-provoking—particularly regarding the multi-dimensional capability evaluation. Due to the inherent differences between agent tasks and traditional LLM tasks, it's difficult to directly design specific test sets for each target capability, as is commonly done in previous LLM benchmarks. So the idea of leveraging graph-based auxiliary information to enable such evaluation is insightful for the community. I hope the authors will open their code to the community.
> >
> > I'd be happy to champion this paper.

---

> > > ### Author Response · Authors · 2025-04-05
> > >
> > > We sincerely appreciate your recognition of our work. Your feedback is truly encouraging and means a great deal to us. We will release the code and dataset upon acceptance of the paper to support progress within the community. Once again, thank you very much for your kind support.

---

### Official Review · Reviewer_DTtP · 2025-03-10

**Overall Recommendation:** 5

**Summary:**

This paper introduces OmniBench, a scalable, graph-based benchmark designed to evaluate multimodal large language model (MLLM)-based virtual agents across multiple dimensions. OmniBench employs a bottom-up subtask composition pipeline to generate 36k tasks with controllable complexity across 20 scenarios. The OmniEval framework introduces Coverage Rate (CR) and Logical Consistency (LC) metrics to assess agents beyond simple success rates. Experiments reveal that current agents, including GPT-4o, struggle with graph-structured tasks, highlighting the need for better reasoning and planning. Fine-tuning on OmniBench improves performance, demonstrating its potential for advancing virtual agent capabilities.

**Claims And Evidence:**

Yes, the claims are supported by clear and convincing evidence. The authors' idea of defining task complexity across five dimensions using five graph properties is impressive.

**Essential References Not Discussed:**

I do not find any essential papers that the authors failed to discuss.

**Experimental Designs Or Analyses:**

Yes, I have checked the experimental designs and analyses. I have reviewed the authors' comparison experiments between OmniEval and human evaluation, the performance of 12 agents on OmniBench, the analysis of task intent's impact on agents, and the comparison between graph-structured and chain-structured tasks.

**Methods And Evaluation Criteria:**

Yes, I think the methods and evaluation criteria make sense. The bottom-up task synthesis approach is highly innovative. More importantly, this synthesis method effectively identifies and reutilizes recurring subtasks in trajectory data, making it highly efficient.

**Other Comments Or Suggestions:**

1. When formatting the paper, pay attention to spacing. For example, the spacing around Section 4 appears too tight and should be adjusted.

2. For tables, the caption is generally placed above.

3. There is a numbering error in the introduction, located in the left section of line 122.

4. The authors should pay attention to some tense errors. There are several instances in the submission where the past tense is incorrectly used.

5. The subfigure in the bottom right corner of Figure 1 seems to be inconsistent with the experimental results in Table 4. The authors should revise it to ensure consistency.

**Other Strengths And Weaknesses:**

### Strengths

**S1.** The authors propose a novel trajectory synthesis method that significantly alleviates the shortage of agent demonstration trajectories. I believe this bottom-up synthesis approach incorporates a divide-and-conquer strategy, overcoming the challenges of directly synthesizing trajectories in a top-down manner by first synthesizing subtask trajectories and then composing them into task trajectories. Additionally, the synthesized trajectories undergo strict data filtering to ensure high quality. Finally, fine-tuning experiments on OmniBench further validate this approach.

**S2.** The authors' Cross-Verification method cleverly integrates subtask trajectory data with evaluation functions. By leveraging mutual verification between these two types of data, it iteratively optimizes both the synthesized trajectories and the evaluation functions simultaneously. This significantly reduces the expert knowledge required to construct evaluation functions for an interactive benchmark.

**S3.** The authors design a graph-based evaluator on a DAG and introduce two novel evaluation metrics, Coverage Rate (CR) and Logical Consistency (LC), enabling a more reasonable and fine-grained evaluation of agents. This evaluation approach differs from traditional result-based and trajectory-based evaluations and is better aligned with real-world tasks that involve complex parallel relationships.

**S4.** The authors define task complexity across five dimensions using five graph properties, enabling the controlled synthesis of tasks with varying complexity. Additionally, these five complexity dimensions are leveraged to design ten agent capabilities through the composition of them. This approach facilitates a multidimensional evaluation of agents, laying the foundation for comprehensive future advancements.

### Weaknesses

I find the ideas in this submission quite intuitive. The bottom-up task synthesis approach and the representation of tasks as graphs are particularly impressive. However, I still have some concerns about this submission:

**W1.** Data Quality

I am somewhat concerned about the quality of the synthesized data. The authors seem to have conducted training experiments only on OmniBench but have not performed similar experiments on other benchmarks. Although Table 3 presents an ablation study demonstrating the effectiveness of their designed quality control module, I would prefer a more intuitive way to showcase this. For example, the authors could evaluate the performance of agents trained on synthesized trajectories on other benchmarks. This might be more convincing than the results in Table 3 or those obtained solely on OmniBench.

**W2.** The Rigor of the Conclusions

Figure 7 shows that agents often struggling to handle graph-structured tasks. And the authors mention on lines 362–365 that "most existing agents are predominantly fine-tuned on chain-structured tasks, which may result in their tendency to interpret graph-structured tasks as linear," attributing this to the chain-structured training data. Does this imply that existing agents plan the next action mainly based on the textual order of the task instructions? I believe the authors' conclusion is important and could offer valuable guidance for the future development of agents, but the authors should conduct relevant experiments to support this conclusion.

**W3.** Experimental Setup

In Table 4, the inputs for MLLMs are all A11Y + Screen, and they perform well under this setting. However, these results cannot be directly compared with those of the agent, as they belong to different experimental settings. I am curious about the performance differences between these MLLMs and specialized agents under a fair comparison on OmniBench.

### Minor Weaknesses

I have some minor comments, which might help improve the writing quality.

1. I am interested in the Cross-Verification algorithm in this submission, which applies to optimizing trajectories and evaluation functions iteratively. But I think the details of this algorithm could be explained more clearly.

2. It would be great to highlight more case studies. Although the authors provide quantitative results in Table 4, qualitative analyses of failure cases are also important for understanding the boundary capabilities of agents.

3. The 20 scenarios mentioned in the submission should be specified in more detail. Additionally, the statistics on OmniBench could be expanded, for example, by including statistics on the synthesized data, such as the step distribution of demonstration trajectories.

**Questions For Authors:**

**Q1.** The authors don't seem to explain the source of the number of test tasks for each capability in the main figure. How were these test tasks obtained? Additionally, were the final test tasks in OmniBench manually filtered?

**Q2.** Are there more examples of input and output resources, and how is their matching ensured so that two subtasks can be seamlessly connected?

**Q3.** In Section 3.2, the authors mention composing a set of predefined APIs into a complete evaluation function using a code LLM (e.g., Claude). I'm curious about the specific APIs included and how the evaluation function is constructed. Could you provide an example of how these APIs are combined to form an evaluation function?

**Relation To Broader Scientific Literature:**

OmniBench builds on recent advancements in graph-based evaluations for agents. The graph-based evaluator in OmniBench is derived from the relevant design in CRAB (CRAB: Cross-environment Agent Benchmark for Multimodal Language Model Agents). Additionally, there have been recent works on synthesized trajectories. For example, AgentTrek (AgentTrek: Agent Trajectory Synthesis via Guiding Replay with Web Tutorials) leverages collected tutorials to synthesize over 10,000 high-quality demonstration trajectories.

**Theoretical Claims:**

The task graph proposed by the authors, combined with the bottom-up theoretical claims, is highly intuitive and I consider them to be correct.

---

> ### Author Rebuttal · Authors · 2025-04-01
>
> We sincerely thank you for professional comments and high appreciation of our work! We are encouraged that our research is recognized as laying the foundation for future advancements. We will address your concerns point by point.
>
> **1.Data Quality**
>
> Thank you for your suggestion, we conduct additional experiments on OmniAct and AndroidControl benchmarks. We follow the experimental setup in the OS-Atlas paper, using our high-quality graph-based trajectories to train our models: **OS-Atlas-4B** and **UGround-7B-V1**. The results are shown in the following tables.
> Our models achieve superior performance on both benchmarks compared to the baselines, which showcases the **effectiveness of our graph-based trajectories**.
>
> **Table:Evaluation on OmniAct**
> |Models|Type-Web|Grounding-Web|SR-Web|Type-Desktop|Grounding-Desktop|SR-Desktop|
> |-|-|-|-|-|-|-|
> |InternVL2-4B|47.51|51.34|24.39|67.00|44.47|29.80|
> |Qwen2-VL-7B|89.22|85.94|78.58|96.27|94.52|91.77|
> |SeeClick|86.98|75.48|68.59|96.79|70.22|72.59|
> |OS-Atlas-4B|88.56|82.00|73.91|96.51|85.53|84.78|
> |UGround-7B-V1|90.16|86.98|79.85|97.13|94.79|91.89|
> |Omni-OS-Atlas-4B(Ours)|89.96|82.74|74.62|97.64|86.37|85.53|
> |Omni-UGround-7B-V1(Ours)|**91.24**|**87.35**|**80.24**|**97.93**|**95.21**|**92.10**|
>
> **Table:Evaluation on AndroidControl**
> |Models|Type-LL|Grounding-LL|SR-LL|Type-HL|Grounding-HL|SR-HL|
> |-|-|-|-|-|-|-|
> |InternVL2-4B|90.94|84.05|80.10|84.09|72.73|66.72|
> |Qwen2-VL-7B|91.94|86.50|82.56|83.83|77.68|69.72|
> |SeeClick|93.00|73.42|75.00|82.94|62.87|59.11|
> |OS-Atlas-4B|91.92|83.76|80.64|84.69|73.79|67.54|
> |UGround-7B-V1|92.15|87.17|83.29|84.72|78.85|70.31|
> |Omni-OS-Atlas-4B(Ours)|**92.49**|83.51|81.38|84.86|73.81|67.71|
> |Omni-UGround-7B-V1(Ours)|92.37|**87.24**|**83.57**|**84.89**|**78.97**|**70.83**|
>
> &nbsp;
>
> **2.The Rigor of the Conclusions**
>
> We appreciate your insightful observations.
> We define the impact of textual order on the model as its instruction sensitivity, conducting experiments with standard deviation as the metric.
> We construct 10 specially designed test tasks, each associated with three task instructions that are semantically identical (based on the same task graph) but differ in textual order.
> As shown in the table below, the original MLLMs tend to be less sensitive to instruction variations, but perform poorly overall.
> Though fine-tuning them on navigation tasks enhances the performance, it also compromises the models' robustness to instructions, with OS-Atlas-Pro and Aguvis exhibiting significantly higher sensitivity.
> Moreover, **after incorporating graph-structured task samples from OmniBench into fine-tuning, the models' performance is further improved while largely preserving their robustness**, with Omni-OS-Atlas and Omni-Aguvis exhibiting reduced sensitivity. This indicates that the trajectory data from OmniBench can help models better recognize complex dependency structures embedded in task instructions.
>
> |Models(backbone)|Avg. Sensitivity|
> |-|-|
> |Human|1.95|
> |InternVL2-4B|2.97|
> |Qwen2-VL-7B|2.58|
> |OS-Atlas-Pro(InternVL2-4B)|9.07|
> |Aguvis(Qwen2-VL-7B)|12.90|
> |Omni-OS-Atlas(InternVL2-4B)|3.49|
> |Omni-Aguvis(Qwen2-VL-7B)|2.67|
>
> &nbsp;
>
> **3.Experimental Setup**
>
> Thank you for your valuable suggestions. Considering MLLMs without grounding-specific training may struggle to perceive fine-grained UI elements in screenshots, we adopt the setup of A11Y+Screen for them. We also conduct the evaluation with only Screenshot setup for fair comparison. Compared to Table4 and the table below, baseline MLLMs consistently achieved poorer performance in Screenshot setup.
>
> |Models|PP|LRP|LSR|LIF|SDM|CDDM|SI|DI|CDK|DSK|
> |-|-|-|-|-|-|-|-|-|-|-|
> |Qwen2-VL-7B-Instruct|3.2|6.4|2.9|3.5|4.7|3.1|4.5|6.7|5.0|6.8|
> |InternVL2-8B|3.3|5.9|2.8|3.2|4.9|3.2|4.7|6.2|4.1|5.3|
> |InternVL2.5-8B|4.9|6.7|4.8|5.7|5.4|6.8|5.2|6.1|5.9|7.2|
>
> &nbsp;
>
> **4.Minor Weaknesses**
>
> For more detailed introduction to Cross-Verification, please refer to Appendix B.2.
> We present additional case studies in Fig1, specify the 20 scenarios in Fig2, and more statistics on OmniBench in Fig3 of the anonymous link: https://anonymous.4open.science/r/OmniBench_rebuttal-A4C7/r3.md
>
> &nbsp;
>
> **5.Questions For Authors**
>
> **To Q1:**
> In OmniBench, test tasks collected from the virtual environment are initially selected based on a joint constraint of five complexity dimensions. We also manually filter the tasks to improve quality.
> **To Q2:** Additional examples of input and output resources are provided in Fig1 via the anonymous link. These resource types, carefully designed by humans, capture the specific states of the virtual environment. Clearly defining these resources for subtasks effectively represents and constrains state transitions in complex virtual environments, facilitating seamless integration between subtasks.
> **To Q3:** We give examples of how APIs are combined in Fig4 of the anonymous link.
>
> &nbsp;
>
> We will integrate these experiments and correct the grammatical errors in the next version of paper.

---

> > ### Comment · Reviewer_DTtP · 2025-04-02
> >
> > Thanks to the authors for their detailed explanations in response to my questions. In the rebuttal, the authors provided clarifications regarding data quality, the rigor of the conclusions, and the experimental setup. I think my concerns have been adequately addressed. I consider OminiBench to be a strong contribution. It has the potential to prompt the community to rethink how to more comprehensively assess the true capabilities of virtual agents. I am willing to raise my score.

---

> > > ### Author Response · Authors · 2025-04-05
> > >
> > > Thank you very much for your strong and encouraging support of OmniBench! We are truly honored that you find our work meaningful and impactful. It is incredibly motivating to know that our contributions resonate with others in the community. Once again, we would like to express our heartfelt thanks for your valuable feedback and strong support.

---

### Official Review · Reviewer_kvCN · 2025-03-12

**Overall Recommendation:** 3

**Summary:**

The paper introduces OmniBench, a scalable, graph-based benchmark designed to evaluate multimodal virtual agents by systematically synthesizing diverse tasks of controllable complexity through automatic task composition. It finds that existing agents significantly struggle with graph-structured tasks compared to linear tasks, and that explicitly including task intents notably improves their performance, especially in long-range planning and decision-making scenarios​

**Claims And Evidence:**

yes

**Essential References Not Discussed:**

no

**Experimental Designs Or Analyses:**

no

**Methods And Evaluation Criteria:**

yes

**Other Comments Or Suggestions:**

1. The graph-based complexity metrics are superficial, e.g. number of nodes and edges, depth and width. More complex correlations between different tasks should also be considered, e.g. the feature correlation between nodes across graph topology [1], or some other metrics summarized in [2]

2. "By incorporating intents into the prompt, we observed a marked improvement in the agents’ performance on tasks designed to evaluate planning (i.e., Long-Range Planning and Parallel Planning)," The improvement brought by idea of intents looks similar as goal-based reinforcement learning.

3. Could you clarify why are the proposed metrics, Coverage Rate (CR) and Logical Consistency (LC), particularly suited for evaluating graph-structured tasks over more traditional metrics such as trajectory similarity or success rate alone?

4. The benchmark utilizes synthesized tasks within controlled environments. Could you discuss how well the proposed evaluation and results can generalize to less structured, real-world virtual agent applications?

5. Could you provide more details on the robustness of intent extraction process, especially regarding the handling of subtasks whose intents might be overlapped or be ambiguously defined?



[1] What Is Missing For Graph Homophily? Disentangling Graph Homophily For Graph Neural Networks. InThe Thirty-eighth Annual Conference on Neural Information Processing Systems.

[2] The heterophilic graph learning handbook: Benchmarks, models, theoretical analysis, applications and challenges. arXiv preprint arXiv:2407.09618. 2024 Jul 12.

**Other Strengths And Weaknesses:**

see below

**Questions For Authors:**

see above

**Relation To Broader Scientific Literature:**

benchmark for virtual agent

**Theoretical Claims:**

no

---

> ### Author Rebuttal · Authors · 2025-04-01
>
> We sincerely appreciate your constructive and insightful comments. We will explain your concerns point by point.
>
> **Q1: More graph-based complexity metrics**
>
> **A1:**
>
> Thank you for the valuable questions. First, we clarify that the current complexity metrics are based on five fundamental graph-based dimensions: dependency, branching, hierarchy, knowledge, and instruction.
> **Despite their simplicity, these node degree-based metrics are effectively aligned with the definition of subtask capabilities.**
> Additionally, following your suggestion, we also adopt feature correlation between nodes as the complexity metrics to conduct further evaluation.
> As the table below shows, both metrics exhibit high Pearson Correlation (ρ) with human evaluations, demonstrating their suitability for accurately reflecting model performance.
> We will include all these metrics in the revision.
>
> |Metrics|ρ(CR, Human)|ρ(LC, Human)|
> |-|-|-|
> |degree-based|0.93|0.95
> |feature correlation|0.94|0.96|
>
> &nbsp;
>
> **Q2: Compared to goal-based reinforcement learning**
>
> **A2:**
>
> Thanks for your insightful observation.
> Both our **intent-based prompting** and **goal-based reinforcement learning** leverage explicit goal representations to guide agents toward long-horizon objectives, thereby improving planning quality. In our case, the intent serves a role similar to goal conditioning in RL: it narrows the agent’s decision space, aligns actions with the final objective, and encourages completion of multi-step plans. This conceptual alignment highlights the value of goal-driven formulations across both reinforcement learning and prompt-based paradigms.
>
> &nbsp;
>
> **Q3: Detailed analysis of CR and LC**
>
> **A3:**
>
> Thanks for your interest in our metric details.
> **(1) Compared to the trajectory similarity:** The trajectory similarity requires strict alignment with a reference path, while graph-structured tasks may have multiple feasible execution paths due to their inherent branching and parallelism. In contrast, **CR** and **LC** are path-agnostic, making them more robust to graph-structured tasks.
> **(2) Compared to the success rate:** The success rate only provides binary judgments of task completion and fails to evaluate the intermediate completion of the subtask. In contrast, **CR** and **LC** can fully leverage the intermediate feedback provided by the graph-based evaluator, allowing for fine-grained evaluation of agent behavior.
> **(3) Further experiments.** We assess Pearson Correlation between human ratings and our metrics as well as traditional metrics on 300 sampled trajectories. The table below shows a **strong alignment** between our metrics and human evaluations, indicating that **CR and LC more faithfully reflect human judgment**.
>
> |Metrics|Pearson Correlation|
> |-|-|
> |Trajectory Similarity|0.52|
> |Success Rate|0.60|
> |CR|0.93|
> |LC|0.95|
>
> &nbsp;
>
> **Q4: Generalization on less structured, real-world virtual agent applications**
>
> **A4:**
>
> Thank you for the valuable question.
> In fact, we have carefully designed realistic tasks in OmniBench, which enable the proposed evaluation and results to effectively generalize to real-world scenarios.
> To validate this, we conduct experiments on OmniAct, which collects data from real devices.
> Compared to Table 4 and the table below, models that performed well on OmniBench also achieve relatively better performance on OmniAct.
> Furthermore, the trajectories from OmniBench also effectively enhance our models' performance in real-world scenarios.
> This indicates that **the evaluation conclusions drawn from OmniBench can be generalized to real-world virtual agent applications**.
>
> |Models|Type-Web|Grounding-Web|SR-Web|Type-Desktop|Grounding-Desktop|SR-Desktop|
> |-|-|-|-|-|-|-|
> |InternVL2-4B|47.51|51.34|24.39|67.00|44.47|29.80|
> |Qwen2-VL-7B|89.22|85.94|78.58|96.27|94.52|91.77|
> |SeeClick|86.98|75.48|68.59|96.79|70.22|72.59|
> |OS-Atlas-4B|88.56|82.00|73.91|96.51|85.53|84.78|
> |UGround-7B-V1|90.16|86.98|79.85|97.13|94.79|91.89|
> |Omni-OS-Atlas-4B(Ours)|89.96|82.74|74.62|97.64|86.37|85.53|
> |Omni-UGround-7B-V1(Ours)|**91.24**|**87.35**|**80.24**|**97.93**|**95.21**| **92.10**|
> More benchmark results are in Fig1 in anonymous link: https://anonymous.4open.science/r/OmniBench_rebuttal-kvCN/r2.md
>
> &nbsp;
>
> **Q5: Robustness of intent extraction process**
>
> **A5:**
>
> Considering the handling of subtasks whose intents might be overlapped or ambiguously defined, we employ an LLM to conduct post-verification processing for the intents.
> It consists of two steps: First, we instruct the LLM to determine whether the extracted intents are clearly and explicitly expressed. If not, we re-extract the intents.
> Second, we maintain a pool of existing task intents to avoid overlap. For each newly extracted intent, the LLM assesses whether it overlaps with any existing intents in the pool. If an overlap is detected, the new intent is discarded.
>
> Thanks again for your valuable suggestions. We will integrate the above content in next version.

---

### Official Review · Reviewer_ASbt · 2025-03-14

**Overall Recommendation:** 3

**Summary:**

This paper introduced OmniBench, a graph-based benchmark that addresses the limitations of existing evaluation frameworks by enabling controllable task complexity through automated subtask composition. The paper also proposes OmniEval, a multidimensional evaluation framework for evaluating virtual agents across 10 capabilities. Evaluation results show that training on this data improves agent generalization.

**Claims And Evidence:**

Claims made in the submission are well supported by experimental results.

**Essential References Not Discussed:**

N/A

**Experimental Designs Or Analyses:**

I have checked the experimental designs and analyses. The experiments are comprehensive. The evaluations on various models reveal performance differences and highlight areas for improvement in current virtual agents, providing valuable feedback for future advancements.

**Methods And Evaluation Criteria:**

The proposed methods and evaluation criteria make sense for the problem.

The OmniBench introduces a graph-based benchmark with automated task synthesis, allowing for controllable complexity through subtask composition. This addresses the issue of uncontrollable task complexity in existing benchmarks.

The OmniEval framework provides a comprehensive evaluation across 10 capabilities, including subtask-level evaluation and graph-based metrics. This multi-dimensional approach offers deeper insights into agent performance compared to traditional evaluation methods.

**Other Comments Or Suggestions:**

N/A

**Other Strengths And Weaknesses:**

Strength:

1. The authors propose an automatic task synthesis method for evaluating virtual agents. This method can significantly improve the diversity and number of evaluation instances and can avoid extensive manual labor.

Weaknesses

1. The authors should provide more detailed examples of the synthesized tasks.

**Questions For Authors:**

N/A

**Relation To Broader Scientific Literature:**

N/A

**Theoretical Claims:**

N/A

---

> ### Author Rebuttal · Authors · 2025-04-01
>
> We greatly appreciate your insightful feedback. To better demonstrate the diversity of automatically synthesized tasks in OmniBench, we restate that our approach first explores a range of various subtasks from the explorable environment and then iteratively synthesizes subtask trajectories and evaluation. Finally, the subtasks are composed into diverse tasks bottom-up.
> Next, we provide more detailed examples of the synthesized tasks.
>
> **Example of the OmniBench Synthesized Tasks**
>
> First, we present 7 subtasks discovered by the MLLM during environment exploration:
>
> |Subtask ID|Instruction Template|Input|Output|Parameter Example|Application|
> |-|-|-|-|-|-|
> |A|Create a new PowerPoint file '{path}'|[]|[ppt_path]|{"path": "./Project_Proposal.ppt"}|PowerPoint|
> |B|Apply the filter '{filter_name}' to the '{path}'|[img_path]|[img_path]|{"filter_name":"Cartoon", "path": "./portrait.png"}|Adobe Photoshop Express|
> |C|Insert the image '{img_path}' into the PowerPoint file '{ppt_path}'|[ppt_path,img_path]|[ppt_path,img_path]|{"img_path": "./scenery.png", "ppt_path": "./Project_Proposal.ppt"}|PowerPoint|
> |D|Download the image from the email sent by '{name}' to '{path}'|[]|[img_path]|{"path": "./portrait.png", "name": "Emily"}|Outlook|
> |E|Send the PowerPoint file '{path}' to '{name}'|[ppt_path]|[ppt_path]|{"path": "./Project_Proposal.ppt", "name": "Emily"}|Outlook|
> |F|Open the local file '{path}' and copy its contents to the clipboard|[]|[text_in_clipboard]|{"path": "./Emily.txt"}|File Explorer|
> |G|Paste the text from the clipboard into the title box on the first slide of the PowerPoint file '{path}'|[text_in_clipboard,ppt_path]|[text_in_clipboard,ppt_path]|{"path": "./Project_Proposal.ppt"}|PowerPoint|
>
> And we introduce 2 tasks constructed bottom-up from the above subtasks:
>
> |Task ID|Instruction|Intent|DAG|
> |-|-|-|-|
> |1|Create a new PowerPoint file named `./Project_Proposal.ppt`, save the image from the email sent by Emily to `./portrait.png` and insert it into the presentation. Then copy the content from the local `./Emily.txt` file into the title box on the first slide. Finally send the PowerPoint file back to Emily.|Create a personal introduction PowerPoint for Emily|{"A":["C"],"D":["C"],"F":["G"],"C":["G"],"G":["E"]}
> |2|Create a new PowerPoint file named `./Project_Proposal.ppt`, insert the image from the email sent by Emily into the presentation, then insert the image with the `Carton` filter applied as well. Finally, send the PowerPoint file back to Emily|Send the comparison PowerPoint to Emily|{"D":["C1"],"A":["C1"],"C1":["C2","B"],"B":["C2"],"C2":["E"]}
>
> We also provide examples of corresponding subtask trajectories and the evaluation function.
>
> **Example of Subtask Trajectory**
>
> ```json
> {
>     "trajectory_id": "XXX",
>     "instruction": "Using the file explorer, navigate to C:\\Users\\user\\Desktop\\images\\ and new a Text Document named introduction.txt",
>     "observations": [
>         obs1.png, obs2.png, ...
>     ],
>     "actions": [
>         {
>             "function": "click_input",
>             "args": {
>                 "button": "left",
>                 "double": false
>             },
>             "rect": [
>                 124,
>                 1020,
>                 179,
>                 1080
>             ],
>             "description": "There are many application icons on the taskbar, and I need to select the File Explorer to complete the task.",
>             "thought": "To fulfill 'Using the file explorer, navigate to C:\\Users\\user\\Desktop\\images\\ and new a Text Document named introduction.txt', I need to first click the 'File Explorer' button to open the corresponding application.",
>             "control_text": "File Explorer"
>         }, ...
>     ],
>     "subtask_id": "XXX"
> }
> ```
>
> **Example of Subtask Evaluation Function**
>
> ```python
> EvalResult = namedtuple('EvalResult', ['success', 'message', 'progress'])
>
> def evaluate_agent_task_completion(dir_path: str, file_name: str) -> EvalResult:
>     # Extract the last directory name
>     dir_path = dir_path.rstrip('/\\')
>     folder_name = os.path.basename(dir_path)
>
>     # Check if navigation to the specified directory was successful
>     if not (check_mouse_clicks(text=folder_name) or check_keyboard_types(text=dir_path)):
>         return EvalResult(False, "Subtask execution fails because agent did not navigate to the specified directory.", 0/2)
>
>     # Check if the new text document was created
>     file_path = os.path.join(dir_path, file_name)
>     if not check_file_exists(file_path=file_path):
>         return EvalResult(False, f"Subtask execution fails because the file was not created in the directory.", 1/2)
>
>     # All checks passed, subtask is considered complete
>     return EvalResult(True, "Subtask completed successfully", 2/2)
> ```
>
> More representative synthesized tasks are visualized in Fig1 of the anonymous link: https://anonymous.4open.science/r/OmniBench_rebuttal-A4C7/r1.md
>
> Thank you again for your suggestion. We will integrate these examples into the next version.

---

### Decision · Program_Chairs · 2025-05-01

**Decision:**

Accept (oral)

**Comment:**

In this work, the authors present OmniBench, a graph-based benchmark designed for evaluating multimodal virtual agents' capabilities of solving complex tasks. The benchmark is synthetically generated, but in a way that the evaluation conclusions drawn from OmniBench can be generalized to real-world virtual agent applications (such as long horizon and graph-structured computer use tasks). In OmniBench, the tasks are generated in a bottom-up fashion: based on a set of synthesized subtasks, the framework composes multiple subtasks in a meaningful way to form a task, and makes sure the resulting task's complexity is on a desirable level.

The work also propose OmniEval, a multidimensional evaluation framework, which uses Coverage Rate (CR) and Logical Consistency (LC) metrics to assess agents beyond success rates.

Reviewers agree that this work's contribution is strong, the benchmark itself as well as the design and procedure to obtain such benchmark are very valuable to the community.
- Citing Reviewer DTtP (score 5): It has the potential to prompt the community to rethink how to more comprehensively assess the true capabilities of virtual agents.
- Citing Reviewer 7v9F (score 4): I find the ideas meaningful and thought-provoking ... the idea of leveraging graph-based auxiliary information to enable such evaluation is insightful for the community... I'd be happy to champion this paper.

The majority of the reviewers' concerns were clarification questions and asking for additional analyses. The authors did a good job addressing the concerns, there are a few additional results that I feel particularly made the work stronger:
- concrete examples of the OmniBench data, including subtasks, tasks, and the evaluation function.
- additional graph-based complexity metrics, and their Pearson correslation with human evaluation.
- results on OmniAct (prior work where data is collected from real devices), providing evidence that the evaluation conclusions drawn from OmniBench can be generalized to real-world virtual agent applications.
- preliminary evidence on the feasibility of using OmniBench data to fine-tuning models.
- additional experiments on incorporating task intent, using both open-source and closed-source models.

Additional AC comments (mostly accessibility improvements):
1. In Section 4.1, the authors could refer to Figure 5 top/mid/bottom rather than just Figure 5, it's a busy figure.
2. L95 left: `Tab. 1` but everywhere else the author use `Table x`
3. The authors could explain what is `A11Y` when they first mention it (L359 right).
4. It seems Appendix C is empty.

Overall this is a strong work. I clearly see its value to present to the ICML community and the broader ML/AI community. I recommend to accept.